# Proteomic and genetic analyses of influenza A viruses identify pan-viral host targets

Influenza A Virus (IAV) is a recurring respiratory virus with limited availability of antiviral therapies. Understanding host proteins essential for IAV infection can identify targets for alternative host-directed therapies (HDTs). Using affinity purification-mass spectrometry and global phosphoproteomic and protein abundance analyses using three IAV strains (pH1N1, H3N2, H5N1) in three human cell types (A549, NHBE, THP-1), we map 332 IAV-human protein-protein interactions and identify 13 IAV-modulated kinases. Whole exome sequencing of patients who experienced severe influenza reveals several genes, including scaffold protein AHNAK, with predicted loss-of-function variants that are also identified in our proteomic analyses. Of our identified host factors, 54 significantly alter IAV infection upon siRNA knockdown, and two factors, AHNAK and coatomer subunit COPB1, are also essential for productive infection by SARS-CoV-2. Finally, 16 compounds targeting our identified host factors suppress IAV replication, with two targeting CDK2 and FLT3 showing pan-antiviral activity across influenza and coronavirus families. This study provides a comprehensive network model of IAV infection in human cells, identifying functional host targets for pan-viral HDT.

Influenza A Virus (IAV) is an enveloped, negative-sense, single-stranded RNA virus that causes mild to severe respiratory disease. Annually in the U.S., the economic burden from IAV is estimated at a total of $11.2 billion[1], with 3–11% of the population experiencing flu symptoms[2]. Symptoms of acute infection include cough, runny nose, fatigue and fever. Disease may progress to acute respiratory distress syndrome or influenza-associated pneumonia in highly susceptible populations, resulting in hospitalization and/or death. IAV circulates yearly as seasonal infections, and zoonotic introduction of new IAV strains can result in a pandemic. For example, influenza A/California/04/2009 H1N1 (pH1N1)-like viruses, the causative agents of the 2009 swine flu pandemic, infected an estimated 11–21% of the world's population in the first year of circulation[3] and caused 200,000 deaths worldwide[4], including many young healthy adults who lacked pre-existing immunity. In comparison, influenza A/Wyoming/03/2003 H3N2 (H3N2), a strain from the 2003–2004 flu season that originally entered the human population in 1968, has average transmissibility but causes increased disease severity[5]. Together, pH1N1-like and H3N2 subtypes are the predominant IAV strains that circulate each season and are

targeted each year by vaccination. Not all zoonotic transmissions result in sustained human-human transmission. For example, influenza A/Vietnam/1203/2004 H5N1 (H5N1) is an avian-derived strain from a 2003–2004 outbreak that caused a small-scale epidemic of human infections after zoonotic transmission from birds, and resulted in severe respiratory disease and >50% mortality rate[6]. While human-to-human transmission is rare, previous studies showed a handful of mutations in H5N1 impart airborne transmissibility in ferrets[7,8], suggesting that H5N1 could mutate to gain transmissibility between mammals and may represent a future risk for pandemic influenza among humans.

Due to antigenic drift, antigenic shift and zoonotic transmission, distinct strains of IAV novel to the human population emerge that have altered transmissibility, pathogenicity and pandemic potential[9,10]. This genetic diversity poses challenges to preventative care and antiviral treatment. Vaccines are developed and administered each year, but have limited and variable effectiveness[11]. There are three classes of approved antiviral therapeutics that target IAV proteins, however increasing prevalence of drug resistance mutations have limited their

✉ e-mail: judd.hultquist@northwestern.edu; robyn.kaake@ucsf.edu; adolfo.garcia-sastre@mssm.edu; nevan.krogan@ucsf.edu

effectiveness, particularly against seasonal IAV strains[12–14]. Moreover, antiviral treatment necessitates very early administration after first symptoms to demonstrate some benefit. IAV's annual burden, potential for future global pandemics and resistance to current treatments highlight a continued need for developing new therapeutics effective against multiple IAV strains.

Host-directed therapies (HDTs) offer an alternative therapeutic approach by targeting host factors essential to virus replication rather than directly targeting viral-encoded factors. Thus, HDTs largely sidestep the challenge of developed drug resistance and have the potential for pan-viral efficacy, as many diverse viruses utilize the same host pathways[15–17]. Proteomic approaches that identify virus-host protein-protein interactions (PPIs) and virus-induced changes in host signaling pathways can pinpoint key host linchpins essential to virus propagation[18–33]. Global IAV-human PPI networks previously generated with lab-adapted IAV strains in immortalized cell lines and in yeast offer good foundations for identifying essential host proteins for IAV infection[24,28–30,34]. However, the overlap between these studies is limited, and most studies relied on cell models that do not represent physiological targets of IAV infection. Combining proteomics with other global approaches such as functional genomics and chemoinformatics can yield actionable HDT targets[21,35]. We and others have demonstrated the utility of a cross-discipline, integrative approach for generating comprehensive models of host reprogramming by a variety of other viral pathogens, and have used these models to identify promising drug candidates[18–20,22,23,36,37].

IAV co-circulates with other respiratory viruses, including severe acute respiratory syndrome coronavirus 2 (SARS-CoV-2), the causative agent of COVID-19 that continues to be a global human health emergency. More transmissible SARS-CoV-2 variants of concern have continued to emerge, many of which have carried resistance mutations to different monoclonal antibody therapeutics[38–42]. It is predicted that SARS-CoV-2 will become endemic and may require regular administration of reformulated vaccine boosters, similar to IAV[43]. Currently, infection by either virus is treated separately by pharmacological or antibody-based antiviral therapeutics approved for clinical use against either SARS-CoV-2 or IAV. Targeting human proteins essential for infection by both viruses could provide pan-respiratory virus HDT.

Here, we employ an integrative systems biology approach to identify human proteins essential for replication across three strains of IAV and SARS-CoV-2. Using affinity purification-mass spectrometry (AP-MS), we identify 332 IAV-human PPIs of pH1N1, H3N2 and H5N1 IAV in three cell types that represent primary and secondary targets of infection. Global proteomic profiling of IAV-infected cells reveals changes in human protein abundance and phosphorylation sites, as well as 13 kinases with changing activity in IAV infection. Whole exome sequencing (WES) data of IAV-infected patients identifies a number of genes with putative loss-of-function (pLOF) variants, including structural scaffold protein AHNAK, that are associated with severe influenza disease and significantly change in our cellular proteomic data. In addition, we perform functional genomic screening of host targets identified in our proteomic dataset and discover 54 human genes that regulate IAV infection, two of which are also important for SARS-CoV-2 replication (*AHNAK* and *COPB1*). Lastly, we test 37 host protein-targeting compounds from our proteomic data and from previously published SARS-CoV-2 phosphorylation data[19] against pH1N1, H3N2 and H5N1 IAV, and find 16 compounds that suppress replication of multiple IAV strains, 5 of which also show antiviral activity against SARS-CoV-2. Collectively, these represent promising antiviral gene targets and potential compounds for future pan-respiratory virus HDT.

## Results
### AP-MS Identifies 332 pH1N1, H3N2 and H5N1 IAV-Human PPIs
We employed a two-pronged proteomics approach and patient exome sequencing to characterize pan-IAV-human protein interactions and

identify putative targets for functional genetic and pharmacological testing (Supplementary Fig. 1). We first performed AP-MS to map PPI networks for pH1N1, H3N2 and H5N1 IAV in A549, NHBE and differentiated THP-1 cells (Fig. 1A). We codon-optimized and cloned 13 2X-Strep-tagged virus proteins (PB2, PB1, N40, PB1-F2, PA, PA-X, HA, NP, NA, M1, M2, NS1, NEP) across the three IAV strains (pH1N1, H3N2, H5N1) (Supplementary Fig. 2A). All 13 proteins are encoded by pH1N1, H3N2, and H5N1 except PB1-F2, which contains a premature stop codon in the pH1N1 viral genome and is not expressed[44,45]. For each cell type after lentiviral transduction, three replicates were treated with universal type I interferon to stimulate an antiviral-like state, and three replicates remained untreated. There were little discernible differences in observed PPIs between interferon-treated and untreated samples, therefore samples were combined totaling six biological replicates (see Methods for details). Eight IAV proteins from all three strains were stably expressed in all three cell types, and all 13 proteins were expressed in at least one cell type for at least one strain, totaling 677 AP-MS samples collected across three IAV strains from three cell types (Supplementary Fig. 2B, Supplementary Fig. 3). Data searched by MaxQuant[46] and scored by Mass Spectrometry Interaction Statistics (MiST)[47] identified 332 total high-confidence PPIs across all strains and human cell types, mapping to a total of 214 human prey proteins (Supplementary Data 1).

Taking the union of all PPIs across the three strains, we identified 111 PPIs from A549 cells, 89 PPIs from NHBE cells and 57 PPIs from differentiated THP-1 cells (Fig. 1B). 29/257 PPIs are shared by at least two of the three cell types, indicating that using multiple cell types substantially expanded the number of PPIs captured to give a comprehensive snapshot of IAV-human interactions. Taking the union of all PPIs across the three cell types, we identified 77 PPIs with pH1N1 among 8 IAV proteins, 77 PPIs with H3N2 among 10 IAV proteins and 142 PPIs with H5N1 among 11 IAV proteins (Fig. 1C). For all three virus strains, NA was expressed at low levels in A549 cells, and not expressed in NHBE or THP-1 cells (Supplementary Fig. 3), therefore no protein interactions passed scoring thresholds. In comparison to a yeast two-hybrid study[34], we observed higher similarity by odds ratio between our study and others that performed AP-MS with exogenously expressed IAV proteins[24,28–30] (Fig. 1D). In total, we discovered 44 novel interactors of IAV that were not found by these previous studies (Supplementary Data 1).

We found a positive correlation between protein sequence similarity and PPI similarity (Fig. 1E), and observed that homologous and non-homologous IAV proteins with high sequence similarity share PPIs, highlighting protein sequence as a driving factor in determining these interactions (Supplementary Fig. 2C, D). For example, N40 is a N-terminal truncation product alternatively translated from the RNA segment encoding PB1, missing only 39 amino acids of PB1[48]. PB1 and N40 share four unique PPIs (36.4% of total unique PB1 PPIs and 50% of total unique N40 PPIs). The functional significance of overlapping PPIs with PB1 is unknown as N40's function is less understood. Between strains, IAV NP has the highest number of shared PPIs (Supplementary Fig. 2E), potentially due to high sequence conservation of NP (Supplementary Fig. 2C)[49,50], and NP's conserved role in viral RNA binding, transcription, trafficking and packaging. pH1N1 NP shares 18 PPIs (85.7% of its total PPIs), H3N2 NP shares 19 PPIs (95% of its total PPIs), and H5N1 NP shares 17 PPIs (56.6% of its total PPIs). Overlap between both homologous and non-homologous IAV proteins is increased when comparing biological pathways among PPIs (Supplementary Fig. 2F). This suggests that while IAV proteins of the different virus strains may target different specific human proteins, they co-opt similar processes or pathways. Gene ontology (GO) enrichment analysis of the PPIs for each IAV protein identified molecular functions previously associated with given IAV proteins (Fig. 1F, Supplementary Data 1). For example, NS1 interactors are enriched for double-stranded RNA binding proteins, consistent with reports showing that NS1 binds

double-stranded RNA to abrogate cellular double-stranded RNA signaling pathways[51,52]. NEP interactors are enriched for actin filament binding, which may expand on NEP's known role in nuclear export of viral RNA[53] and could indicate a novel function for NEP in post-export association with and trafficking of viral RNA along cytoskeleton filaments[54]. Enrichment terms also characterize IAV proteins of unknown function, such as heat shock protein binding and chaperone binding for N40, which may indicate a novel role for N40 in protein translation and/or stability. PB2 and HA have no significant enrichments that passed our thresholds, due to the small number of PPIs (Fig. 1C, Supplementary Data 1). However, M1, which also has a small number of PPIs, showed significant enrichment in PPIs with translational elongation factor activity. M1 known functions include facilitating nuclear export and trafficking of viral RNA[55,56]; it is unclear if these PPIs are involved in this activity or indicate an independent, novel function for M1.

### IAV PPI networks from three cell types identify strain-specific and pan-IAV-human interactions

The collective 332 high-confidence PPIs include interactions between: 108 human proteins and nine IAV proteins in A549 cells; 88 human proteins and eight IAV proteins in NHBE cells; and 56 human proteins and eight IAV proteins in THP-1 cells (Supplementary Fig. 4). Interactions shared across multiple cell types include cleavage and

polyadenylation factor (CPSF) complex members that interact with NS1 in all three cell types, spliceosome components that interact with NP in NHBE and THP-1 cells, and mitochondrial ribosome subunits that interact with NP in NHBE and THP-1 cells (Supplementary Fig. 4). These complexes have known roles in IAV infection, such as NS1 interacting with the CPSF complex to post-transcriptionally dampen host mRNA expression and innate immune response[57,58]. While the majority of the IAV-human PPIs appear cell type-specific, it should be noted that some viral proteins did not express consistently in all cell types (Supplementary Fig. 2B, Supplementary Fig. 3), and some high-confidence PPIs identified in one cell type were also identified in other cell types but below our stringent scoring thresholds (Supplementary Data 1).

Noting the small PPI overlap between cell types (Fig. 1B), we reasoned the three cell types collectively provide a representative snapshot of IAV-human interactions. We therefore took the union of all PPIs across the three cell-type specific networks to generate one unified interactome to visualize pan-IAV and strain-specific interactions between 214 human proteins and 12 IAV proteins (Fig. 2). Interactions shared by all three strains are represented by tricolored nodes and include proteins involved in: the spliceosome (NP, 11 interactors), CDC5L complex (M2, 3 interactors), mitochondrial ribosome (NP, 17 interactors), 60 S ribosome (PA, 3 interactors), nuclear transport (M2, 10 interactors), macroautophagy (M2 and PB1-F2, 2 interactors), and proton transport (M2, 2 interactors) (Fig. 2). These pan-IAV PPIs may

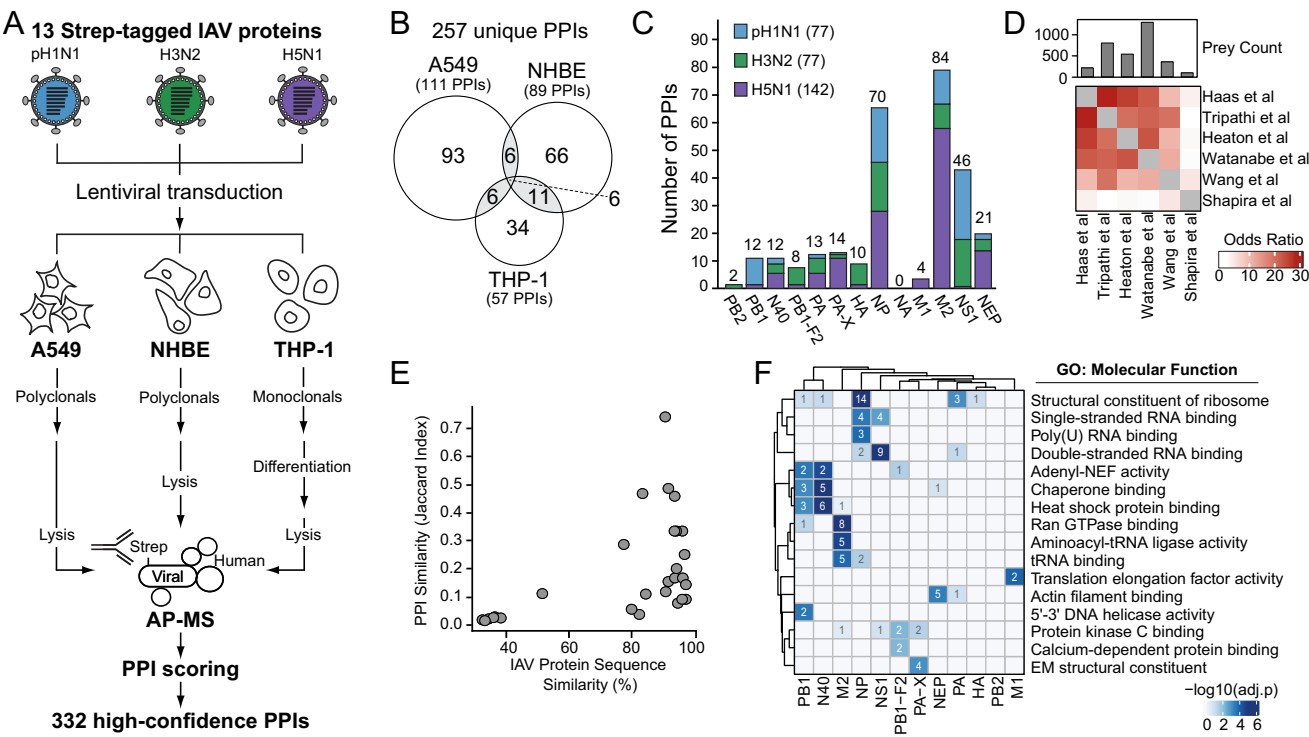

**Fig. 1 | AP-MS identifies 332 pH1N1, H3N2 and H5N1 IAV-human PPIs. A** AP-MS experiment design. 13 2X-Strep-tagged proteins from pH1N1, H3N2 and H5N1 IAV were individually transduced by lentivirus to generate stable A549, NHBE and THP-1 cell lines. A549 and NHBE cells were cultured as polyclonal pools. THP-1 cells were cultured as monoclonal isolates and subsequently treated with Phorbol-12-myristate-13-acetate (PMA) to induce differentiation into a macrophage-like state. All cells were treated with doxycycline to induce IAV protein expression for 24 hours and subsequently lysed. Affinity-purified IAV proteins and co-purified human proteins were identified by MS and scored to assign interaction confidence. **B** Venn diagram of unique IAV-human PPIs identified in each cell type. The total 332 high-confidence PPIs were unified across virus strains, resulting in 257 unique PPIs by cell type and 29 PPIs that are shared in at least two of the three cell types (grey shading). **C** Bar graph of the unique IAV-human PPIs identified for each IAV protein and strain. PPI numbers reported are unified across cell types. **D** Identification correlation

matrix comparing the human interacting proteins identified by AP-MS in this study with other published studies that used AP-MS with affinity-tagged IAV proteins exogenously expressed in cell lines[28,30], AP-MS in the context of virus infection[24,29], and an orthologous yeast two-hybrid approach[34]. **E** Comparison of shared protein interactions (PPI similarity) by Jaccard index against IAV protein sequence similarity. PPIs reported are unified across cell types. **F** Heatmap of gene ontology (GO) molecular function enrichments among the human interacting proteins of indicated IAV proteins, unified across all strains and cell types and clustered by correlation of enrichment profiles. GO terms were curated from the top 3 nonredundant terms with at least 2 genes for at least one IAV protein. Increasing shading intensity reflects increasing significance of the enrichment term. Number of proteins per enriched cluster are shown in white if significant (adjusted *p*-value < 0.002; one-sided Fisher's exact test), and grey if not significant (adjusted *p*-value > 0.002; one-sided Fisher's exact test).

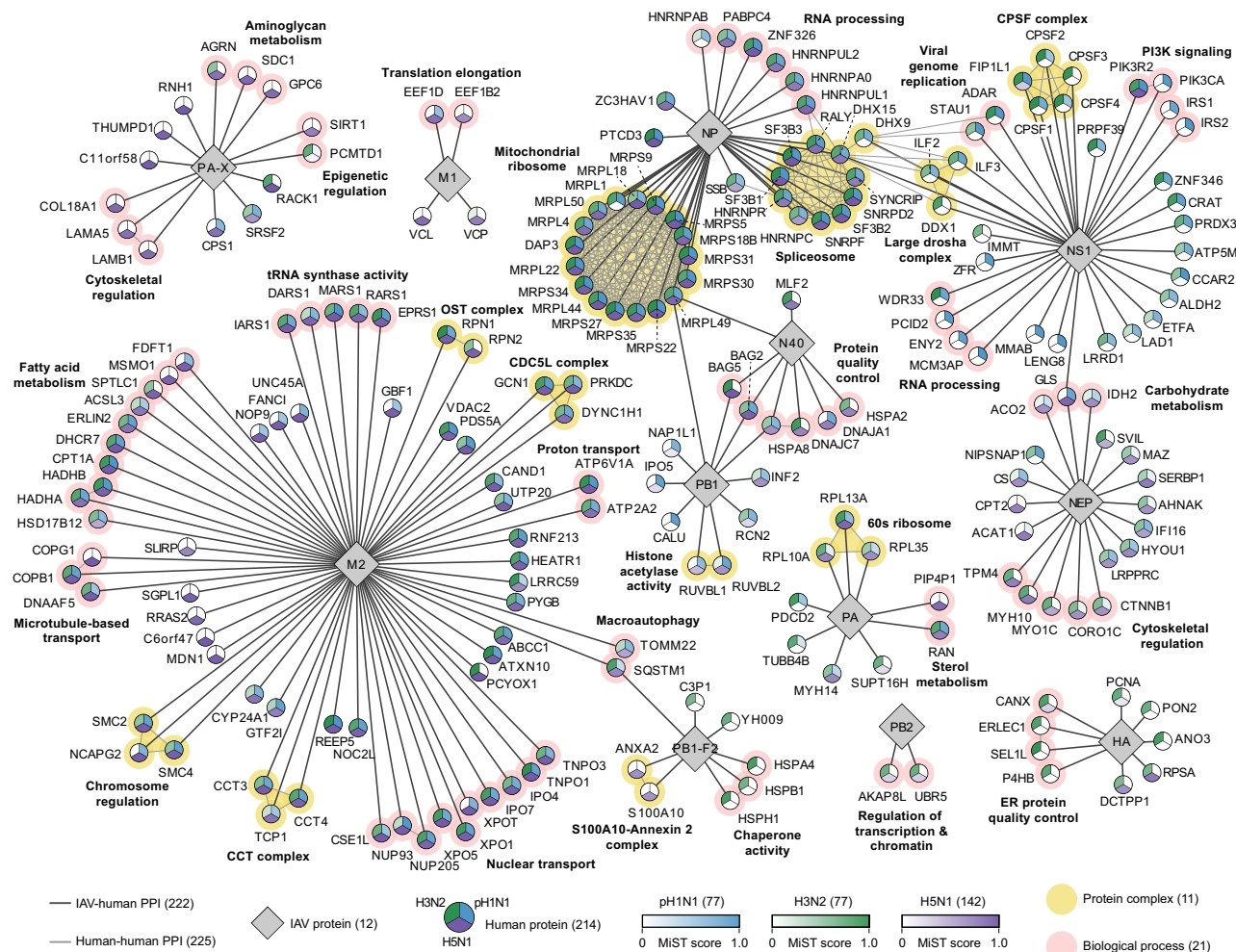

**Fig. 2 | IAV PPI networks from three cell types identify strain-specific and pan-IAV-human interactions.** High-confidence IAV-human PPIs between 12 IAV proteins (grey diamonds) and 214 human proteins (circular nodes) identified from three IAV strains unified across the three cell types. Human protein nodes are split into three sections and colored by the IAV strain for which the interaction was identified: pH1N1 (blue), H3N2 (green) and H5N1 (purple). Color shading is proportional to MiST PPI confidence score (scale at bottom; not identified represented by white color), enabling visualization of high-confidence interactions that scored above our MiST score thresholds and interactions with additional IAV strain(s) detected in our AP-MS data that fell below our MiST score thresholds. For PPIs that are shared between multiple IAV proteins or cell types, the maximum MiST score from either IAV protein or cell type is reported in the network for each strain. IAV-human PPIs are depicted (dark grey lines), and human-human PPIs are identified (light grey lines) as curated in the CORUM[130] database. Human protein complexes (yellow halo) are labeled as described in CORUM[130], and biological processes (pink halo) are labeled as described by GO terms.

indicate high functional importance for infection, and their corresponding biological processes are consistent with known essential roles of the interacting IAV proteins in viral genome replication and translation (NP, PA) and viral assembly or entry and budding (M2)[59]. The interactome also highlights strain-specific PPIs represented by nodes with one or two colors, which are most noticeable among NS1, PA-X and PB1-F2, viral proteins largely involved in host response (Fig. 2). These may represent unique co-opting of host protein complexes by each strain. For example, NS1 interactions are predominantly identified with pH1N1 and H3N2, including four PI3K signaling components, consistent with prior data showing that NS1 activates PI3K signaling during infection to modulate host apoptotic response[60,61]. While H5N1 NS1 sufficiently expressed in THP-1 cells, H5N1 NS1 had low expression by Western blot in A549 and NHBE cells (Supplementary Fig. 3). Consequently, most interactions for H5N1 NS1 did not pass stringent scoring thresholds (Supplementary Data 1), with the exception of PIK3R2. Likewise, H3N2 PB1-F2 interactions are largely involved in protein chaperone activity, while H5N1 PB1-F2 interactions are part of the S100A10-Annexin 2 protein complex that has roles in membrane trafficking, connecting cytoskeletal components to the cell membrane,

and cell adhesion[62]. It is unknown if the PB1-F2 interactions identified in our network contribute to its known activities in apoptosis, innate immune response, and IAV virulence[63–65], or suggest potential novel cellular roles.

Some IAV-interacting host proteins in our network have been previously reported (Fig. 1D, Supplementary Data 1), which lends confidence to our network. For example, vacuolar ATPase catalytic subunit component ATP6V1A is functionally important for IAV entry[66,67]. In our study, ATP6V1A interacts with M2 from all three IAV strains in all three cell types (Fig. 2, Supplementary Fig. 4). ATP6V1A had the highest MiST interaction confidence score with H5N1 M2 in all three cell types (Supplementary Data 1), therefore we validated this interaction by endogenous reciprocal IP in H5N1-infected A549 cells. Using parallel reaction monitoring (PRM) MS, we show that H5N1 M2 pulls down with ATP6V1A and is enriched (log2 fold change 1.2) compared to an IgG control pulldown in H5N1-infected A549 cells (Supplementary Fig. 5A, B, Supplementary Data 1). In addition to known interactions, our IAV-host network also identified novel PPIs, such as NEP interactors involved in cytoskeletal regulation and RNA binding, including structural scaffold protein AHNAK (Fig. 2). AHNAK interacts

with H5N1 NEP in NHBE cells (Supplementary Fig. 4, Supplementary Data 1), and reciprocal pulldown of endogenous AHNAK in H5N1-infected NHBE cells co-purified NEP with an enrichment of log2 fold change 2 compared to an IgG1 control pulldown in H5N1-infected NHBE cells (Supplementary Fig. 5C, D, Supplementary Data 1). We also demonstrate that AHNAK and NEP both localize to the cytoplasm in IAV-infected NHBE cells (Supplementary Fig. 5E). While HA is reported to interact with CANX for proper HA folding and processing at the ER[24,68], three HA PPIs with ER protein quality control machinery in our network (ERLEC1, SEL1L, P4HB) are novel and may identify additional human proteins involved in HA folding and processing. These are specific to H3N2 HA and A549 cells in our network (Supplementary Fig. 4, Supplementary Data 1), but it should be noted that pH1N1 HA did not express in A549 and HA from all strains did not express in NHBE or THP-1 cells (Supplementary Fig. 3). Lastly, we identified eight high-confidence human protein interactors of N40, including six that are involved in protein quality control machinery, indicating a potential novel cellular role for under-characterized N40 in modulating human and/or viral protein expression. Collectively, the interactome highlights the rich biology of human proteins and pathways targeted by three strains of IAV.

### Global proteomic profiling highlights 13 modulated kinases in IAV infection

In an orthogonal proteomic approach, we performed global protein abundance and phosphorylation profiling on pH1N1, H3N2 or H5N1 IAV-infected primary NHBE and differentiated THP-1 cells at four time points post-infection to identify IAV-modulated host signaling pathways (Fig. 3A). MS data searched by MaxQuant[46] and quantified by MSstats[69] identified hundreds of significant protein abundance changes and site-specific phosphorylation events that occur over the time course of infection for each IAV strain in each cell type (Fig. 3B, C, Supplementary Data 2). We detected increasing IAV NP abundance across the collected time points, indicating productive infection, though this rise varied slightly by strain (Fig. 3D). Not all time points passed MS quality control (e.g. 12 hours in THP-1 cells), and we therefore selected a single time point representing peak IAV infection for all subsequent analyses where NP abundance reached comparable high levels across the strains (Fig. 3D). Moderate overlap in phosphorylation events was observed between the three strains, with the seasonal circulating IAV strains (pH1N1 and H3N2) sharing a larger overlap with each other than with avian-derived H5N1 (Supplementary Fig. 5F). Proteins with increased phosphorylation in at least one site across all strains were functionally enriched in RNA splicing and processing, cellular and nuclear membranes, regulation of gene silencing, and innate immune response (Supplementary Fig. 5G, Supplementary Data 2), consistent with IAV co-opting host RNA machinery to splice and translate viral RNA, IAV entry and exit, viral-induced gene silencing by blocking nuclear export of host mRNA, and cellular detection and response to virus[59,70].

Since protein phosphorylation changes reflect changes in kinase activities, we next leveraged our phosphoproteomic data to predict kinases with altered activity during IAV infection. We observed a weak correlation between protein and phosphorylation site abundance changes, suggesting that the observed phosphorylation changes are largely not driven or biased by changes in protein abundance (Fig. 3E, Supplementary Fig. 5H). Using a comprehensive catalog to map kinase-substrate relationships[71] with substrate proteins identified in our phosphorylation data and in-house scoring criteria to increase confidence of kinase activity annotations (Methods), we identified 13 kinases with activity changes during IAV infection (Fig. 3F, Supplementary Data 2). In NHBE and THP-1 cells, five mitogen-activated protein kinase (MAPK) family members (MAP2K3, MAP2K6, MAP-KAPK2, MAPKAPK3, MAPKAPK5) showed decreased activity or no significant change during pH1N1 infection, and increased activity

during H3N2 and H5N1 infection (Fig. 3F). This may be directly related to the differential capacity of NS1 from different strains to activate JNK and PI3K signaling[72]. This trend is also observed for two ribosomal protein S6 kinase (RPS6K) family members (RPS6KB1, RPS6KB2) (Fig. 3F), although the functional significance is unclear. In THP-1 cells, one member of the phosphatidylinositol 3-kinase-related kinase family (PRKDC) showed increased activity during pH1N1 and H3N2 infection, and no significant change during H5N1 infection (Fig. 3F). This may indicate a macrophage-specific response via PRKDC with the predominant human-infecting IAV strains. Collectively, these predictions identify differential kinase activity patterns during infection, and may be indicative of the different IAV strain pathogenicities and host responses.

Overlaying PPI and phosphoproteomics results identified 45 human proteins with IAV-modulated phosphorylation sites that interact with at least one IAV protein (Fig. 3G). For some PPIs, the phosphoregulation pattern is consistent across all three strains and may represent pan-IAV functionality in infection and host response. For example, CANX, an H3N2 HA interactor, was downregulated in phosphorylation at Serine 583 (S583) by all three IAV strains in both THP-1 and NHBE cells. SRSF2, a pH1N1 PA-X interactor, was universally upregulated at S208 by all three IAV strains. For other PPIs, phosphorylation is differentially regulated by IAV strain and may represent strain-specific regulation of protein activity or localization. One example is CPSF4, a well-known NS1 interactor that blocks nuclear export of host pre-mRNA and post-transcriptionally inhibits the production of interferon-stimulated genes as part of NS1-mediated host cell shutoff[57,58,73,74]. Here, CPSF4 S200 phosphorylation was regulated in a strain-specific manner, with decreased phosphorylation during pH1N1 and H3N2 infection and increased phosphorylation during H5N1 infection (Fig. 3G). Strain-specific differences in the functionality of CPSF4-NS1 interaction have been reported, namely pH1N1 NS1 is unable to block mRNA export and stimulate mRNA translation as efficiently as H5N1 subtype virus[73,74]. While most PPIs were mapped with one or two IAV-modulated phosphorylation sites, five PPIs had four or more IAV-regulated phosphorylation sites (Fig. 3G). These include DNA damage-sensing kinase PRKDC, an M2 PPI that was upregulated in activity during infection (Fig. 3F) and had five upregulated phosphorylation sites upon pH1N1, H3N2 and H5N1 infection (Fig. 3G); and AHNAK, a large (~700 kDa) scaffold protein and an NEP PPI that had 11 IAV-modulated phosphorylation sites (Fig. 3G). The additional layer of phosphoregulation during IAV infection for IAV-human PPIs, including PRKDC and AHNAK, may highlight increased functional importance of the interaction in infection.

### Patient exome sequencing identifies gene variants encoding proteins that are regulated in AB and PH during IAV infection

To investigate the clinical implications of our proteomic datasets, we explored the correlation between proteins we found to be regulated in cell models of IAV infection and patient responses to IAV. To this end, we obtained de-identified human blood samples following informed consent from individuals at five eMERGE study sites. We used principal components analysis (PCA) to characterize the genetic ancestry of the study population and identified 495 individuals of genetically-identified European Ancestry, of whom 161 were hospitalized with severe influenza infection and 334 served as outpatient controls (Fig. 4A). We used International Classification of Diseases, Ninth Revision (ICD-9) codes as the phenotypic trait for the analysis. Whole exome sequencing of the 495 participants achieved 97% coverage of targeted bases at a depth of ≥ 20x. In total across all participants, we identified 3,621,267 variants in 22 million base pairs across the coding regions of 22,621 genes, of which 90% were rare variants (minor allele frequency (MAF) < 1%). There was no evidence of site-specific effects or other systematic biases in the analysis of the filtered data.

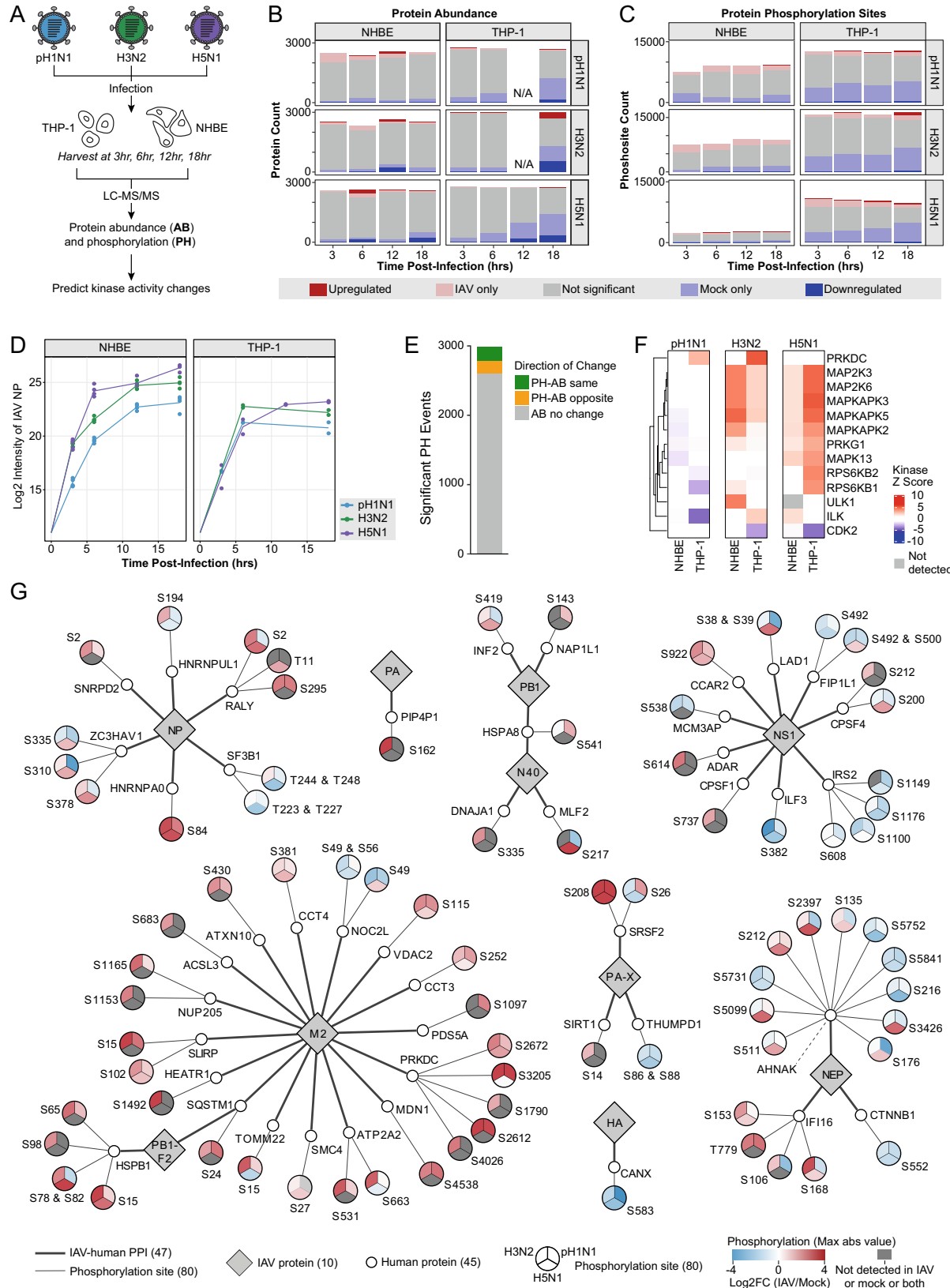

In order to investigate whether proteins from our proteomics study impinged on genes with variants associated with severe influenza disease, we searched for predicted loss-of-function (pLOF) variants in our proteomic datasets. We focused on pLOF variants as these are not only easier to predict[75–77], but have more understandable molecular consequences than gain-of-function or synonymous mutations. We identified 196,832 total variants in genes corresponding to the 3658 AB, 3656 PH, and 214 PPI proteins that we detected in our proteomic datasets (Supplementary Data 3). Variants were classified as pLOF if identified as nonsynonymous exonic, frameshift substitution or stop gain/loss (MAF < 1%), and predicted deleterious from any of six annotation algorithms (Supplementary Data 3). In total, 2808 AB

**Fig. 3 | Global proteomic profiling highlights 13 modulated kinases in IAV infection. A** Experimental design workflow for global proteomic profiling of protein abundance (AB) and phosphorylation (PH) changes in NHBE and PMA-differentiated THP-1 cells infected in biological duplicate with pH1N1, H3N2 or H5N1 IAV (MOI 2, four time points post-infection with time-matched mocks). **B–D** AB data are not available (N/A) for pH1N1 and H3N2 IAV at the 12-hour time point in THP-1 cells, as these samples did not pass MS quality control. Bar chart plotting the total number of (**B**) proteins from the AB dataset and (**C**) phosphorylation sites from the PH dataset quantified at each time point (light red=significantly increased; light blue=significantly decreased; dark red=only detected in IAV infection; dark blue=only detected in mock infection; grey=no significant change). **D** Log2 intensity of IAV NP AB detected over the time course of pH1N1, H3N2 and H5N1 IAV infection in NHBE and THP-1 cells. **E–G** All represented data corresponds to 18 hours (pH1N1, H3N2) and 12 hours (H5N1) post-IAV infection. **E** Correlation of the PH and AB data

for all peptides where significant changes in both protein PH and AB could be measured (green=PH-AB change in the same direction; yellow=PH-AB change in the opposite direction; grey=no significant AB changes). Correlation data is represented as a total across all virus strains and cell types. **F** Heatmap of predicted kinase activity (kinase Z score) with FDR < 0.05 from IAV-infected NHBE and THP-1 cells (red=increased activity; blue=decreased activity; grey=not detected). **G** IAV-human PPI map of 10 IAV proteins (grey diamonds) interacting with 45 human proteins (small white circles) that possess significantly changing phosphorylation sites (adjusted *p*-value < 0.05; two-sided *t*-test). Significantly changing phosphorylation sites (emanating large circular nodes) are stratified by IAV strain (pie sections) and colored by the maximum log2 fold change (log2FC) (IAV/mock; red=increase, blue=decrease, grey=not detected). Phosphorylation sites detected across multiple cell lines are represented by the maximum absolute value, non-infinite fold change.

genes, 3092 PH genes, and 177 PPI genes had pLOF variants, however, since the power to detect singletons is limited by their low frequency, we used a gene-based collapsing method by which rare mutations are considered jointly for association analysis[78]. After gene-based collapsing, 1082 AB genes, 2336 PH genes, and 118 PPI genes were tested for association with severe influenza disease (Fig. 4B). We identified 24 AB, 49 PH and 5 PPI significant severe disease-associated genes with pLOF variants (FDR < 0.05) (Supplementary Data 3). For further analyses between genes with pLOF variants and our proteomic dataset, we considered 95 AB, 161 PH and 7 PPI genes as moderately significant genes with pLOF variants associated with severe disease (FDR < 0.1) to include more genes with smaller effects (Fig. 4B–D). Looking at the overlap of our proteomic and patient datasets, we found 23 AB and 52 PH genes with pLOF variants to be significantly regulated during IAV infection (Fig. 4C, E, F). From the list of phosphorylation sites identified in our proteomic data, we also identified pLOF nonsynonymous mutations at predicted phospho-serine, phospho-threonine, and phospho-tyrosine sites for each of the AB, PH and PPI datasets (Supplementary Data 3). We identified phospho-variants in 75 AB genes, 146 PH genes and 6 PPI genes (Supplementary Data 3).

At the convergence of our proteomic and pLOF analyses are 44 proteins with pLOF variants that were detected in all three proteomic datasets: AB, PH, and PPI (Fig. 4D). Focusing on these proteomically detected genes with pLOF variants and looking at the average of the severe influenza disease association FDR across the three dataset tests, we find two surpassing our significance threshold (FDR < 0.1): *AHNAK* and *SEL1L*. SEL1L interacts with H3N2 HA, and while detected in both AB and PH datasets, it is not significantly regulated during IAV infection at the protein level. In contrast, we find that AHNAK, a H5N1 NEP interactor, is significantly downregulated in protein AB by all three IAV strains in both NHBE and THP1 cells (Fig. 4E), and has significant changes in phosphorylation at a number of phosphorylation sites in at least one cell type by at least one IAV strain (Fig. 4F). Looking at all AHNAK predicted phosphorylation sites, as well as all of the significantly regulated phosphorylation sites in our dataset (including infinite quantification values) (Fig. 4G), we found serine position 210 contained phosphorylation disruption mutations of serine to proline or glutamine (S210P/Q) that was significantly associated with severe disease (FDR < 0.05) (Fig. 4H, Supplementary Data 3). Interestingly, AHNAK phosphorylation at S210 was also regulated during infection of Calu-3 cells with early-lineage or Alpha variant SARS-CoV-2[79,80]. Together, our data pinpoints how molecular regulation of AHNAK in response to infection could be reflective of the systemic effect of AHNAK as it relates to disease severity in the host.

### siRNA knockdown identifies 54 pro-viral and antiviral factors of IAV and SARS-CoV-2 infection

To functionally validate PPI and kinase factors in IAV infection, we adapted an arrayed siRNA screening approach[81] in A549 cells to identify host proteins whose knockdown suppressed infection (pro-viral

factors) or enhanced infection (antiviral factors) (Fig. 5A). A total of 290 genes were knocked down in biological duplicate, and include: (1) 212/214 IAV interacting proteins that were targetable by siRNA; and (2) a panel of 64 kinases (including 12 kinases from Fig. 3F) and 14 phosphorylated proteins (Supplementary Data 4). Knockdown cells were immunostained for IAV NP and quantified by flow cytometry for percent NP-positive (%NP + ) cells as a readout for percent IAV infection (Supplementary Fig. 6A). We then calculated the log2 fold change in % NP-positive (NP + ) cells of experimental siRNA against the mean of multiple non-targeting (NT) control siRNA. To assess if siRNA knockdown affected viability, we performed cell viability staining which showed siRNA knockdown cells were above 92% viable (Supplementary Data 4). Since siRNA knockdown alone did not meaningfully reduce cell viability, we next asked if cell viability resulting from the combination of siRNA knockdown and IAV infection biased observed changes in IAV infection. Increased or decreased IAV infection was not correlated with increased or decreased viability of cells with siRNA knockdown and IAV infection (Fig. 5B), therefore no gene knockdowns were removed from analysis due to toxicity. The two replicates showed a good correlation ($R^2 = 0.78$) for log2 fold change in IAV infection (Fig. 5C). As expected, NT control siRNA did not affect IAV infection (black dots, Fig. 5C), and IAV NP-targeting siRNA inhibited IAV infection (green dots, Fig. 5C).

We classified pro-viral and antiviral factors using a threshold log2 fold change of ≤ -2 or ≥ 2, respectively, for the IAV PPI and PH screens (Fig. 5D, E, Supplementary Data 4). Using this cutoff for the PPI screen, we classified 44 genes as regulators of IAV infection, of which 37 were pro-viral and 7 were antiviral factors (Fig. 5D). These 44 functional proteins interact with 12 IAV proteins, corresponding to at least one functional interaction per IAV protein. In comparison to previous genome-wide siRNA knockdown studies[66,82–84], we identified 37 novel human proteins that functionally affect IAV infection. Whereas the referenced genome-wide screens reported a < 2% hit rate for identifying genes that functionally affect IAV infection, our AP-MS-based strategy achieved a higher hit rate (20.6%) for identifying functional nodes. This is consistent with previous findings that genetic screens based on PPI data show ten-fold higher hit rates for identifying functional factors of infection by IAV and other viruses[30,85]. From the PH targets, 10 were classified as pro-viral factors and include two IAV-regulated phosphoproteins and eight kinases (Fig. 5E, Supplementary Data 4). Collectively among both screens, 47 host factors that regulated IAV infection are pro-viral, highlighting the strength of proteomics-based technologies in identifying human protein nodes critical to infection.

SARS-CoV-2 is a respiratory RNA virus that infects similar cell types as IAV and may target similar host proteins for infection, therefore we endeavored to identify human proteins essential for both viruses. We knocked down the 54 functional IAV host factors identified above and challenged these cells with SARS-CoV-2. Cells were assessed for viability with siRNA knockdown by

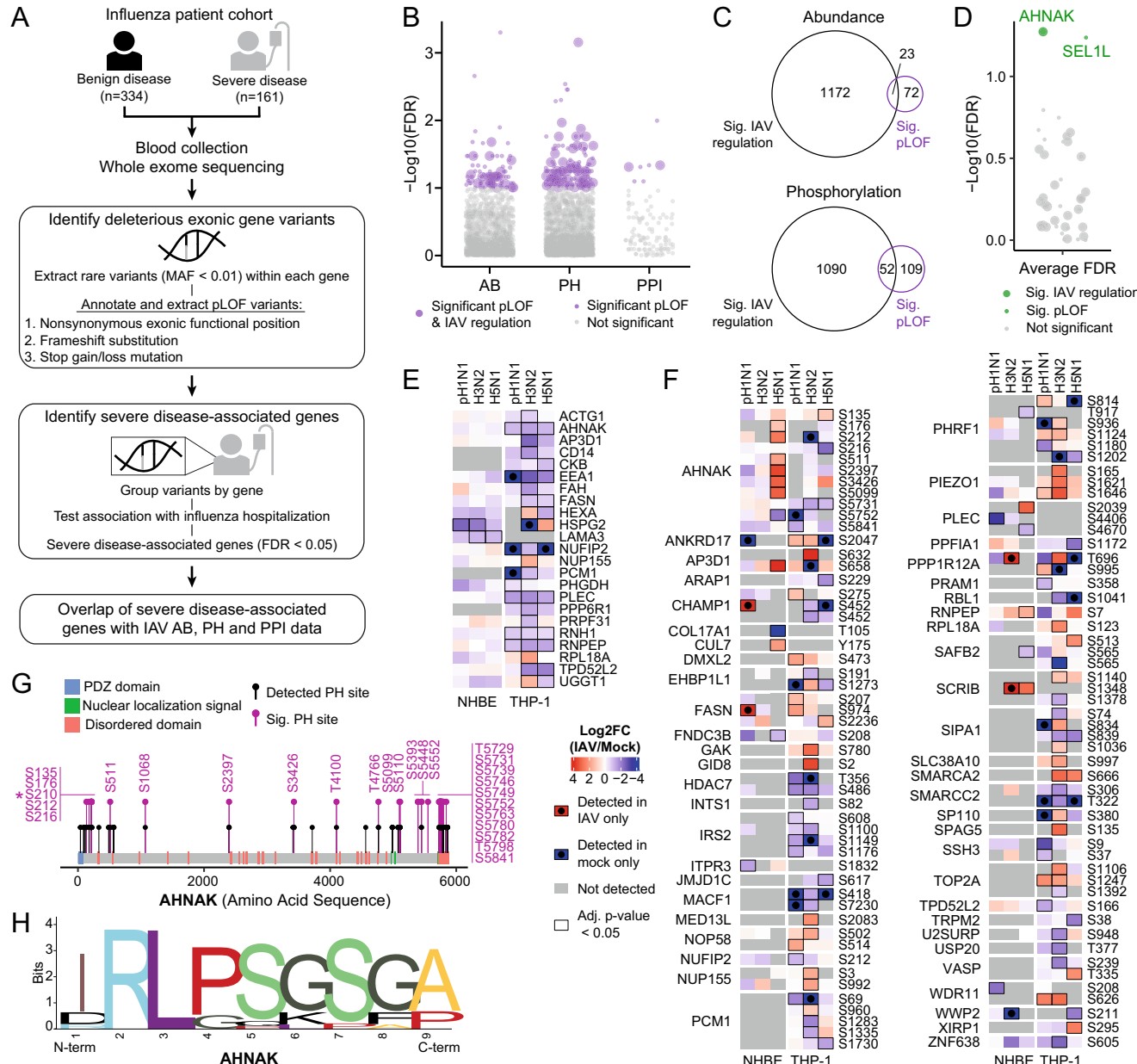

**Fig. 4 | Patient exome sequencing identifies gene variants encoding proteins that are regulated in AB and PH during IAV infection. A** Schematic representation of sample collection and data analysis for identifying genes with pLOF variants associated with severe influenza disease from an influenza patient cohort. Genes with pLOF variants plotted against the false discovery rate (-log10(FDR)) from the severe-disease association test for (**B**) each of the AB, PH and PPI datasets or (**D**) the average FDR across the three AB, PH, and PPI datasets (purple and green circles=genes with significant pLOF variants (FDR < 0.1); grey=genes with pLOF variants below threshold (FDR > 0.1); large circles=genes with significant protein AB or PH changes (adjusted *p*-value < 0.05; two-sided *t*-test); small circles=genes detected in AB or PH proteomic datasets with no significant changes). **C** Venn diagram of the overlap of proteomic datasets with significant changes in AB (top, left) or PH (bottom, left), and significant genes with pLOF variants (corresponds to the total number of purple circles in **B**). Heatmap of log2FC in infection vs mock (log2FC

(IAV/Mock)) from NHBE and THP-1 cells at 18 hours (pH1N1, H3N2) and 12 hours (H5N1) post-IAV infection (reported in Supplementary Data 2) for (**E**) AB of 23 significantly changing proteins, and (**F**) PH of 52 changing phosphorylated proteins, that have significant pLOF variants (from the union in **C**) (red=increase, blue=decrease, grey=not detected; red box with black circle=only detected in IAV infection, blue box with black circle=only detected in mock infection; black box outline=significant change (adjusted *p*-value < 0.05; two-sided *t*-test). **G** AHNAK phosphorylation sites detected and significantly changed in the PH data (black pins=detected no significant change; pink pins=significantly changed (adjusted *p*-value < 0.05; two-sided *t*-test); asterisk=AHNAK S210). **H** Multiple sequence alignment sequence LOGO (S210P/Q, middle) for phosphorylation disruption mutations in AHNAK created with WebLogo version 2.8.2[148] using motifs identified by pLOF analysis as likely to be loss of phosphorylation.

cytotox staining and infected with SARS-CoV-2 at MOI 0.1 for 72 hours, and SARS-CoV-2 infection was quantified by RT-qPCR against viral N protein. Cell viability staining showed siRNA knockdown cells had a median cell viability ranging 94.3-99.1% of all siRNA knockdown cells across the replicates (Supplementary Data 4). We calculated the log2 fold change of SARS-CoV-2

infectivity in knockdown cells for experimental siRNA against replicate-matched NT siRNA, and classified pro-viral and antiviral factors using a threshold of median log2 fold change ≤ -2 or ≥ 2, respectively. This identified three IAV PPI factors that regulated SARS-CoV-2 infection: COPB1, AHNAK and RUVBL2 (Fig. 5F, Supplementary Data 4). While we report these three proteins as IAV

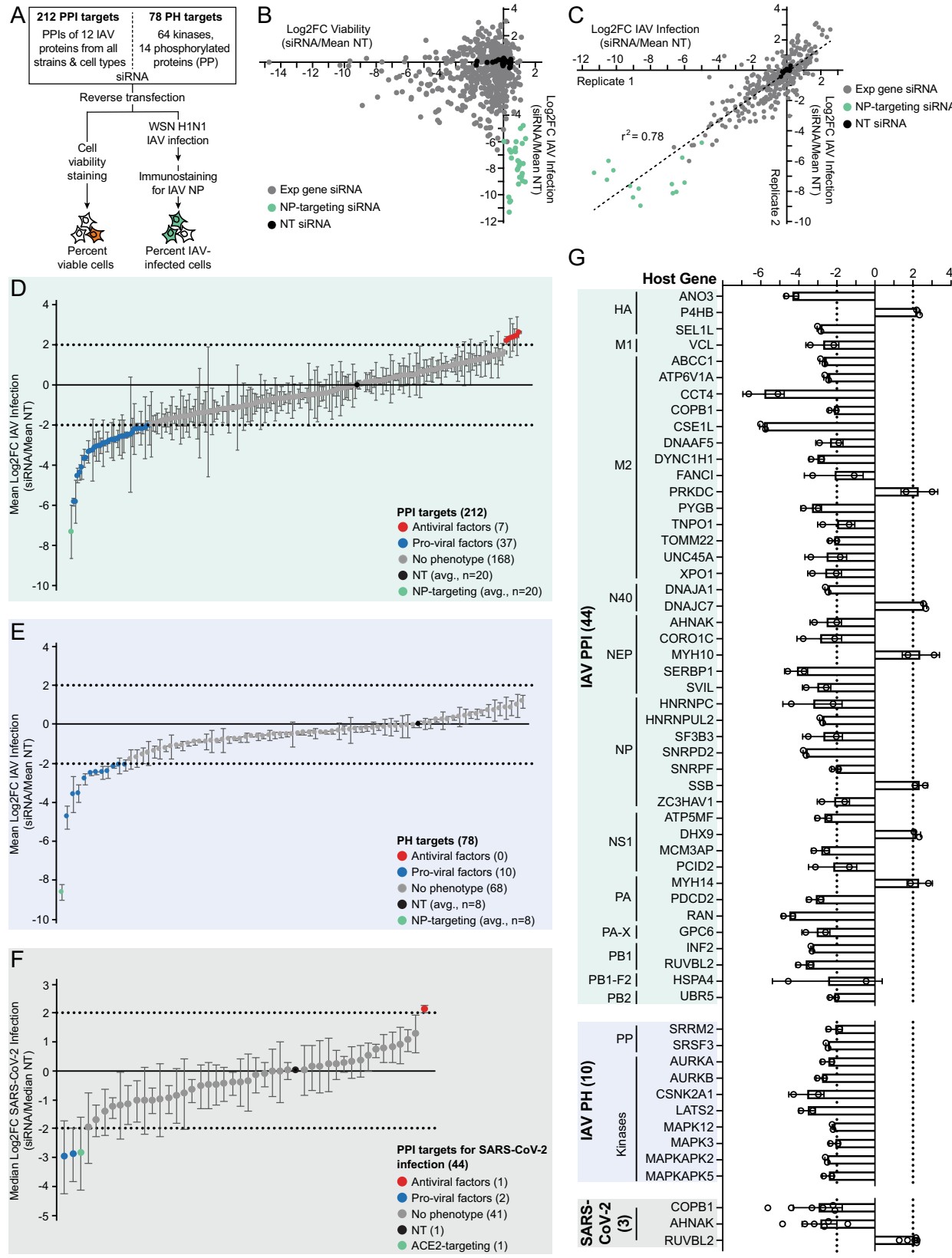

PPIs, to our knowledge, none are previously reported as PPIs with SARS-CoV-2 proteins. AHNAK was profiled as an RNA binding protein whose RNA binding kinetics peak early in SARS-CoV-2 infection[86]. No siRNA knockdowns from the PH dataset passed our log2 fold change thresholding criteria for SARS-CoV-2

(Supplementary Fig. 6B), however, 7 out of 10 PH targets mildly decreased SARS-CoV-2 infection with significance ($p$-value < 0.02) (Supplementary Data 4). Collectively, we classified 54 regulators of IAV infection, three of which act as pan-respiratory virus regulators of IAV and SARS-CoV-2 infection (Fig. 5G).

**Fig. 5 | siRNA knockdown identifies 54 pro-viral and antiviral factors of IAV and SARS-CoV-2 infection. A** Arrayed siRNA screen approach in A549 cells reverse-transfected in $n = 2$ biologically independent samples with gene-targeting, non-targeting (NT) or IAV NP-targeting siRNA, and infected with Influenza A/WSN/1933 H1N1 (MOI 0.1, 24 hours). Cell viability (live-cell staining) and percent IAV infection (%NP-positive (%NP + ) cells; immunostaining for IAV NP) were quantified by flow cytometry. Correlation plots comparing: **B** the cell viability against IAV infection for each siRNA from each biological duplicate; or **C** the variation in IAV infection between the biological duplicates. Log2FC was calculated by normalizing %viable or %NP+ cells for each siRNA against the mean of multiple replicate-matched NT siRNA (siRNA/mean NT) (green dots=IAV NP-targeting siRNA, black dots=NT siRNA, grey dots=experimental gene-targeting siRNA). Distribution of log2FC in IAV infection for (**D**) 212 PPI targets and (**E**) 78 PH targets, plotted as the mean of $n = 2$ biologically independent samples per target. The log2FC in IAV infection was calculated for each siRNA against the mean replicate-matched NT siRNA (blue dots=pro-viral factors (mean log2FC < -2), red dots=antiviral factors (mean log2FC > 2), grey dots=no/weak phenotype; green dot=IAV NP-targeting siRNA; black dot=NT siRNA; error bars represent standard deviation). **F** Distribution of log2FC in SARS-CoV-2 infection for 44 IAV PPI targets plotted as the median of six replicates ($n = 2$ biologically independent samples, each in $n = 3$ technical replicates) per target. The log2FC in SARS-CoV-2 infection was calculated for each siRNA against a replicate-matched NT siRNA (blue dots=pro-viral factors (median log2FC < -2), red dots=antiviral factors (median log2FC > 2), grey dots=no/weak phenotype; green dot=ACE2-targeting siRNA; black dot=NT siRNA; error bars represent median absolute deviations (MAD)). **G** Bar chart of pro-viral and antiviral factors for IAV and SARS-CoV-2 screens plotted as the mean log2FC in IAV infection (data re-plotted from **D** and **E**; error bars represent standard deviation) and the median log2FC in SARS-CoV-2 infection (data re-plotted from **F**; error bars represent MAD).

## Host-directed compounds targeting IAV and SARS-CoV-2 host factors identify inhibitors of pH1N1, H3N2 and H5N1 IAV infection

To identify potential HDT against IAV infection, we screened compounds targeting a subset of the 44 siRNA-validated pro-viral and antiviral PPI factors and 13 kinases with IAV-modulated activity changes (Fig. 6A). 20 host proteins were targetable by at least one compound, and include 8 PPIs (targeted by total 16 compounds) and 12 kinases (targeted by total 15 compounds), with two host proteins identified as both PPIs and kinases (total 29 unique compounds). Overlaying our phosphorylation data of IAV infection in human NHBE and THP-1 cells with phosphorylation data of SARS-CoV-2 infection in Vero E6 cells[19] and human lung epithelial Calu-3 cells[79], we noted eleven shared kinases (Supplementary Fig. 6C) with similar predicted kinase activity profiles upon infection by both IAV (Supplementary Data 2) and SARS-CoV-2[19,79] (Supplementary Fig. 6D). For example, MAPK signaling members (MAP2K3, MAP2K6, MAPKAPK3, MAPKAPK5, MAPKAPK2, MAPK13) and RPS6K signaling members (RPS6KB1 and RPS6B2) showed increased activity, and CDK2 showed deceased activity (Supplementary Fig. 6D). Therefore, we leveraged known kinase-targeting SARS-CoV-2 antiviral compounds[19] and additionally tested these for potential dual activity against IAV (Fig. 6A). A total of 37 unique host-directed compounds were screened against pH1N1, H3N2 and H5N1 IAV infection in A549 cells (Supplementary Fig. 7, Supplementary Data 5). In total, we identified 16 compounds with antiviral activity against at least one IAV strain, with 7 compounds showing pan-IAV antiviral activity (Supplementary Data 5).

Four compounds targeting four PPIs show antiviral activity against at least two IAV strains (Fig. 6B–E). Bafilomycin A1 which targets M2 PPI V-ATPase subunit ATP6V1A, daunorubicin which inhibits M2 PPI ATP-binding cassette subfamily member ABCC1, and PACMA31 which targets HA PPI protein disulfide-isomerase P4HB showed potent pan-IAV antiviral activity (Fig. 6B, C, E). Bafilomycin A1 has also been reported to inhibit IAV infection with PR8 H1N1 in A549 cells[87]. DNA-dependent protein kinase (DNA-PK) inhibitor NU7441 targeting M2 PPI DNA-PK PRKDC suppressed pH1N1 and H3N2 infection but not H5N1 infection (Fig. 6D). Interestingly, this strain specificity is also reflected in the phosphoproteomic data, as PRKDC showed increased kinase activity in pH1N1 and H3N2 infection but not H5N1 infection (Fig. 3F). To our knowledge, daunorubicin (ABCC1) and NU7441 (PRKDC) are novel inhibitors of IAV infection.

In addition to the PRKDC kinase inhibitor, we found inhibitors of five additional kinases showed antiviral activity against IAV (Fig. 6F–H, Supplementary Data 5). Dinaciclib, an inhibitor of cyclin-dependent kinase CDK2, showed potent antiviral activity against all three strains (Fig. 6F). A previous study reported dinaciclib antiviral activity with H7N9 IAV[88], further supporting broad spectrum potency. MRT68921, an inhibitor of autophagy-activating kinase ULK1, showed antiviral activity against H3N2 and H5N1 (SI > 2), and decreased pH1N1 infection (SI < 2) (Fig. 6G, Supplementary Data 5). ULK1 in complex with other proteins activates mTOR-dependent autophagy[89], a pathway that is necessary for IAV infection[90]. To our knowledge, MRT68921 is a novel antiviral for IAV, likely acting through ULK1 inhibition to downregulate autophagy and suppress infection. Additionally, three inhibitors of four members of the mitogen-activated protein kinase (MAPK) pathway showed antiviral activity (Fig. 6H). Lestaurtinib, which targets MAP2K3 and MAP2K6, showed antiviral activity against pH1N1 and H5N1 (SI > 2), and decreased H3N2 infection (SI < 2) (Fig. 6H). Interestingly, while MAP2K3 and MAP2K6 show increased predicted kinase activity in H3N2 and H5N1 IAV infection but not pH1N1 IAV infection (Fig. 3F), Lestaurtinib inhibits pH1N1 IAV infection (Fig. 6H). MAPK-13-IN-1, which targets MAPK13 (p38δ), showed broad spectrum activity with some differences in potency between the three IAV strains (Fig. 6H). Although SI values for MAPK-13-IN-1 could not be quantitatively calculated based on the concentrations we used, the lack of toxicity at the tested concentrations indicates SI is likely to be above 2 (Supplementary Data 5). PF-3644022, which targets MAPKAPK2, showed antiviral activity against H5N1, though was not tested against pH1N1 (Fig. 6H). Taken together, these three MAPK-targeting compounds suggest the MAPK signaling pathway may be essential for multiple strains of IAV infection and targetable for host-directed antiviral therapy.

Three out of the eight SARS-CoV−2 antiviral compounds showed antiviral activity against at least two strains of IAV (Fig. 6I–K). Gilteritinib, which targets AXL kinase functioning upstream of p38, MAP2K3 and MAP2K6, showed antiviral activity against the three IAV strains tested (Fig. 6I). While inhibitors against MAP2K3 and MAP2K6 (lestaurtinib) and MAPK13 (p38δ) (MAPK13-IN-1) showed antiviral activity against multiple IAV strains (Fig. 6H), two SARS-CoV-2 antiviral p38 inhibitors had no effect on IAV infection (ralimetinib targeting MAPK14 (p38α) and MAPK11 (p38β); ARRY-797 targeting MAPK14 (p38α)) (Supplementary Fig. 7, Supplementary Data 5). This suggests that IAV and SARS-CoV-2 converge on upstream kinases in the MAPK pathway, and that their kinase signaling activity is essential for infection. Pictilisib targeting PIK3CA and PIK3CD also showed broad spectrum antiviral activity with strain-specific differences in potency (Fig. 6J). SI values for pictilisib could not be quantitatively calculated based on the concentrations we used, however the lack of toxicity indicates SI is likely to be above 2 (Supplementary Data 5). Previous findings that other PIK3CA-targeting compounds inhibit infection by two different IAV strains support PIK3CA as a targetable node for IAV treatment[91]. PIK3CA has been shown to act through PR8 H1N1 IAV protein NS1[61], and we identified PIK3CA as an interactor of pH1N1 NS1 (Fig. 2), but how PIK3CA regulates H3N2 and H5N1 infection is unclear. MK-2206, which targets the AKT kinases (AKT1, AKT2, AKT3), showed strong antiviral activity against H5N1 and moderate antiviral activity against pH1N1 (Fig. 6K). AKT signaling may be a unique host signaling pathway more heavily utilized by H5N1 avian-derived IAV strain. In addition to

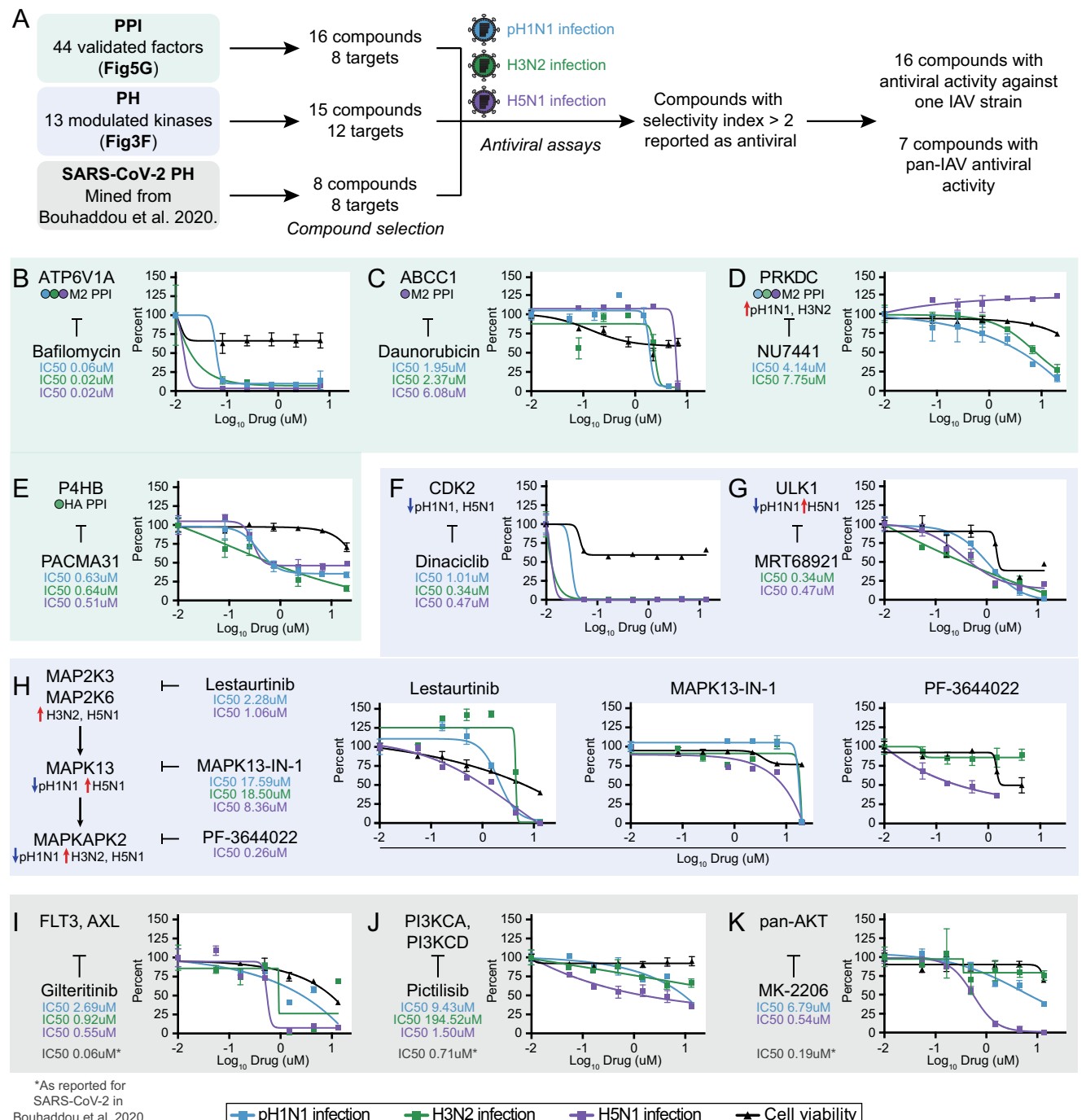

**Fig. 6 | Host-directed compounds targeting IAV and SARS-CoV-2 host factors identify inhibitors of pH1N1, H3N2 and H5N1 IAV infection. A** Compounds targeting eight IAV PPI factors and 12 IAV-modulated kinases were manually curated by literature search and selected based on target specificity and drug availability. 8 kinase-targeting compounds with antiviral activity against SARS-CoV-2[19] were included. In total, 37 unique compounds were screened against pH1N1, H3N2 and H5N1 IAV infection. Compounds with selectivity index (SI) [CC50/IC50] > 2 were classified as having antiviral activity (Supplementary Data 5, see also Source Data). **B–K** A549 cells were pre-treated with compound at the indicated doses (2 hr) and infected with pH1N1 (MOI 0.5), H3N2 (MOI 0.5) or H5N1 (MOI 0.05) IAV for 24 hr. Percent IAV-infected cells were quantified by immunostaining for IAV NP followed by high throughput imaging (blue line=pH1N1; green line=H3N2; purple line=H5N1). Percent alive cells were quantified by MTT assay in uninfected A549 cells (black line). Data points represent the mean across *n* = 3 biologically independent samples. Schematics mark the target with corresponding PPI or PH dataset and IAV strain, and the corresponding compound (at left). Compounds are annotated with

IC50 values for IAV strains in which SI > 2. Error bars represent standard error of mean (SEM). **B–D** Dose-response curves for M2 PPI-targeting compounds, including: ATP6V1A-targeting compound bafilomycin A1; ABCC1-targeting compound daunorubicin; and PRKDC-targeting compound NU7441. PRKDC is also a kinase identified in the IAV PH data. **E** Dose-response curve for HA PPI P4HB-targeting compound PACMA31. **F–G** Dose-response curves for PH kinase-targeting compounds, including: CDK2-targeting compound dinaciclib; and ULK1-targeting compound MRT68921. **H** Dose-response curves for members of the MAPK pathway (pathway schematic at left), including MAP2K3, MAP2K6, MAPK13 and MAPKAPK2, each annotated with corresponding compounds. PF-3644022 was not tested against pH1N1. **I–K** Dose-response curves for SARS-CoV-2-mined antiviral compounds targeting three kinase pathways: FLT3 and AXL targeted by gilteritinib; PI3KCA and PI3KCD targeted by pictilisib; and AKT1, AKT2 and AKT3 (pan-AKT) targeted by MK-2206. SARS-CoV-2 IC50 values are included as previously reported[19], where SARS-CoV-2 infection was quantified by RT-qPCR of SARS-CoV-2 N protein in compound-treated A549-ACE2 cells.

gilteritinib, pictilisib and MK-2206, pan-IAV compounds dinaciclib (targeting CDK2) (Fig. 6F) and MAPK13-IN-1 (targeting MAPK13) (Fig. 6H) are reported to have antiviral activity against SARS-CoV-2[19], representing a total of five compounds with pan-antiviral activity across IAV and SARS-CoV−2.

## Discussion

This study represents an integrative systems biology approach that unifies cellular proteomic data with patient genomic data to generate a comprehensive network model of IAV infection. By studying functional host factors of circulating or zoonotic IAV strains, we identified essential, druggable host targets that may serve as potential treatment strategy alternatives to increasingly obsolete classes of IAV protein-targeting drugs. Using a two-pronged proteomic approach, we interrogated three different IAV strains (pH1N1, H3N2, and H5N1) in multiple cells types of infection (primary bronchial epithelial, lung epithelial and myeloid cell lines) and identified novel strain-specific and pan-IAV PPIs and IAV-modulated host kinase pathways. Combining the cellular proteomic data with whole exome sequencing data from an influenza patient cohort, we pinpointed a number of potential molecular regulators of host response and determinants of disease outcome. By functional genetic screening, we found 54 human genes that map back to 44 PPI factors and 10 PH factors act as pro-viral and antiviral factors in IAV infection. Three of these IAV host factors also regulate infection by SARS-CoV-2, acting as pro-viral (COPB1, AHNAK) or antiviral (RUVBL2) factors of SARS-CoV-2 infection. Screening compounds that target IAV-interacting and IAV-modulated proteins identified 16 compounds that suppress replication of at least one strain of IAV, with seven compounds exhibiting pan-IAV activity and five compounds inhibiting multiple strains of IAV and SARS-CoV-2. While our study focused on identifying promising antiviral targets for potential pan-viral HDT in future influenza and COVID-19 treatments, we recognize there is more to be mined from our data, especially in teasing apart strain- or cell-type specific interactions and their consequence on different disease prognoses or outcomes.

Three human protein interactors of two IAV proteins targeted in antiviral drug development (M2, HA) show promising evidence as broad spectrum HDT targets. These three host proteins functionally affect IAV infection (Fig. 5G), and compounds targeting these three host factors show pan-IAV antiviral activity: daunorubicin targeting ABCC1 (M2 interactor); NU7441 targeting PRKDC (M2 interactor and IAV-modulated kinase); and PACMA31 targeting P4HB (HA interactor) (Fig. 6C–E). M2 is the IAV protein target of amantadine and rimantadine inhibitors, two classes of antivirals approved for clinical use that are now obsolete for IAV treatment due to virus resistance, particularly among 2009 pH1N1-like and H3N2 seasonal strains[92,93]. HA is one of the IAV proteins responsible for host cell entry by binding to sialic acid on epithelium cells, and is an attractive but challenging target for antiviral therapeutics due to high antigenic drift and shift[94,95]. There are no HA-targeting drugs currently available for clinical use. Recent antiviral strategies instead target human sialic acid to block HA binding and IAV entry. For example, sialic acid inhibitors were recently shown to target HA and have antiviral efficacy[96]. Antiviral compound Fludase (DAS181), a bacteria-derived sialidase fusion protein that cleaves sialic acid from epithelium cell surface to prevent IAV entry[97], enrolled its first patient in a Phase III clinical trial in 2019 and is moving towards FDA approval and clinical use. Fludase is a promising example of a host-directed strategy to successfully treat IAV infection. Here, we identified three host-directed pre-clinical and FDA-approved compounds that target two functional IAV M2 PPIs and one HA PPI, all of which have antiviral activity against pH1N1, H3N2 and H5N1 IAV infection. None of these compounds are currently in clinical trials or approved for use in treating influenza. Collectively, these three compounds represent potential alternative, host-directed targets for treating influenza

disease. As with all host-directed therapies, the potential toxicity associated with their use needs to be carefully addressed.

IAV co-circulates seasonally with SARS-CoV-2 and other respiratory pathogens, which presents a significant challenge for public health. In cell models and in mice, IAV infection led to increased susceptibility to SARS-CoV-2 co-infection, increased SARS-CoV-2 viral loads, and resulted in more severe lung damage, morbidity and mortality[98,99]. This observation is specific to IAV in comparison to co-infection with other respiratory viruses[98]. IAV and SARS-CoV-2 present similar respiratory disease symptoms, and current antiviral treatment is specific to either virus. Here, we have taken a novel approach that combines genetic and pharmacological screening to identify host node vulnerabilities of both IAV and SARS-CoV-2 for potential pan-respiratory virus HDT. We challenged the 54 functional IAV PPI and PH factors by siRNA knockdown against SARS-CoV-2 infection, and identified two human genes, COPB1 and AHNAK, that act as pro-viral factors in both IAV and SARS-CoV-2 infection (Fig. 5G). A third gene, HNRNPUL2, acts as a pro-viral factor in IAV infection (Fig. 5G) and falls just below our log2 fold change cutoffs as a pro-viral factor for SARS-CoV-2 infection (Supplementary Data 4). While we report these factors as IAV-human PPIs (M2-COPB1, NEP-AHNAK), to our knowledge, neither are reported as SARS-CoV-2-human PPIs. COPB1, a subunit of the coatomer complex I (COPI) that is associated with non-clathrin coated vesicles and involved in endosomal transport, is essential for IAV infection in other siRNA-based studies[82,83,100], although to our knowledge its interaction with IAV M2 is novel. COPB1 was also shown to be required for infection of other RNA viruses including vesicular stomatitis virus[101], and the secretory pathway was shown to promote SARS-CoV viral RNA replication and synthesis[102]. COPB1 and its role in endosomal transport may facilitate essential steps in viral RNA synthesis and trafficking or viral assembly for IAV and SARS-CoV-2. HNRNPUL2 interacts with IAV NP in our study, and is reported to interact with SARS-CoV-2 nucleocapsid (N) protein[103]. IAV NP and SARS-CoV-2 N proteins share functional similarities; both proteins are viral RNA (vRNA)-binding proteins involved in encapsidation of vRNA and formation of viral ribonucleoproteins (vRNPs), vRNP trafficking, and virus replication[59,104]. NP and N proteins are abundantly expressed during infection, and evolutionarily conserved among related influenza viruses and coronaviruses, respectively, which make them attractive broad spectrum drug or vaccine targets[104–107]. HNRNPUL2, a predominantly nuclear protein with RNA binding activity whose cellular function is under-characterized, likely facilitates NP and N in essential vRNA replication or vRNP trafficking functions. While we report these three factors as promising candidates for pan-viral HDT, future work is needed to determine the specific mechanisms by which these proteins affect IAV and SARS-CoV-2 infection.

Out of eight compounds that show antiviral activity against SARS-CoV-2[19] and that target kinase pathways detected in our IAV phosphorylation data, three compounds show antiviral activity against at least two IAV strains: gilteritinib, pictilisib and MK-2206 (Fig. 6I–K). As mentioned above, two additional compounds identified from our IAV PH data, dinaciclib and MAPK13-IN-1, show antiviral activity against all three strains of IAV in our study (Fig. 6F,H), and act also as antivirals against SARS-CoV-2 as reported in a previous study[19]. In total, five compounds from our study show antiviral activity against multiple strains of IAV and SARS-CoV-2 infection. This highlights the power of leveraging and mining orthogonal phosphoproteomic analyses of infection by different respiratory viruses to identify novel pan-viral HDT. These five compounds target kinases of diverse pathways, and include FLT3/AXL (gilteritinib), MAPK (MAPK13-IN-1), PI3K (pictilisib), AKT (MK-2206) and CDK (dinaciclib) signaling pathways (Supplementary Data 5). None of these compounds are currently in clinical trials for influenza or COVID-19. While our results with pictilisib (targeting PI3KCA, PI3KCD) and MK-2206 (targeting AKT1, AKT2, AKT3) are novel for IAV, other PI3K and AKT signaling inhibitors are in clinical

trials for influenza[108]. Collectively, these findings represent novel potential pan-respiratory antiviral HDT.

Several protein targets are identified by multiple orthogonal data in this study and warrant further investigation. One example is AHNAK, a large, ~700 kDa structural scaffold protein that interacts with H5N1 NEP above our PPI scoring thresholds, and with pH1N1 and H3N2 NEP below our thresholds (Fig. 2, Supplementary Data 1), indicating the interaction may not be strain-specific. In our network, the AHNAK-NEP interaction is specific to NHBE cells (Supplementary Fig. 4, Supplementary Data 1) and could be a result of cell type-specificities, but may also result from experimental or protein expression and purification differences of NEP in the other cell types (Supplementary Fig. 3). Reciprocal pulldown of endogenous AHNAK co-purified H5N1 NEP in H5N1-infected NHBE cells, and both AHNAK and NEP are localized to the cytoplasm of NHBE cells during productive H5N1 IAV infection (Supplementary Fig. 5C–E). AHNAK has 11 sites that are differentially regulated in phosphorylation with pH1N1, H3N2 and H5N1 infection, with about half of the sites universally up- or downregulated and half of the sites regulated in strain-specific patterns (Fig. 3G). Interestingly, *AHNAK* is identified as a significant gene with pLOF variants associated with patients who experienced severe influenza disease (Fig. 4D), regulated in protein AB and PH during cellular IAV infection (Fig. 4E, F), and contained phosphorylation disruption mutations at site serine 210 significantly associated with severe influenza disease (Fig. 4H), indicating AHNAK may play an important role in disease outcome. *AHNAK* is also a pan-respiratory virus gene target, as its knockdown decreases both IAV and SARS-CoV-2 infection (Fig. 5G). AHNAK's role in IAV and SARS-CoV-2 infection is unknown, but may be tied to viral RNA export and/or virus budding. Calcium-dependent cell-cell contact formation has been shown to trigger AHNAK's phosphorylation by protein kinase B and relocalization outside the nucleus[109], localization to the plasma membrane and complex formation with S100A10-Annexin 2 complex[110]. AHNAK has been proposed to coordinate cytoskeleton and membrane architecture changes together with S100A10-Annexin 2 complex[110–112]. This function is important in pathogen infection with bacterium *Salmonella*, where AHNAK is recruited to membrane ruffles and is required for infection[112]. IAV NEP facilitates export of viral ribonucleoprotein (vRNP) complexes from the nucleus to the cytoplasm[53,113], and facilitates virus formation and budding at the plasma membrane through its interaction with a membrane-embedded F-type proton-translocating ATPase[114]. AHNAK and IAV NEP may coordinate cellular cytoskeletal and membrane remodeling for vRNP export and trafficking or IAV assembly and budding at the membrane. We found five additional human proteins involved in cytoskeletal regulation interact with NEP (Fig. 2), and NEP PPIs are enriched in actin binding (Fig. 1F), which support this model. To date, no published studies show AHNAK as an interactor of SARS-CoV-2 proteins, however AHNAK was profiled as an RNA binding protein whose RNA binding kinetics peak early in SARS-CoV-2 infection[86]. Our study uniquely identifies AHNAK as essential for SARS-CoV-2 infection, perhaps through AHNAK's interaction with viral RNA that may play a critical role in viral RNA production, trafficking or assembly during infection.

In summary, this study highlights the unique strength of an integrative systems biology approach to generate multi-dimensional data profiling IAV, and identify functional and druggable human proteins essential for IAV infection. By utilizing AP-MS, global proteomics, patient exome sequencing, functional genetics and pharmacological screening, we identify human gene targets and compounds that can be a starting point to develop potential pan-IAV and pan-respiratory viral HDT (Figs. 5G and 6B–K). We hope the highly collaborative approach to data-driven target identification for host-directed therapies presented here can be employed to find additional pan-viral therapies and mechanisms beyond IAV and SARS-CoV-2 for other infectious diseases.

## Methods

### IAV-human PPI AP-MS methods

**IAV strep-tagged plasmid and lentivirus construction.** The coding sequences of 12 virus proteins for A/California/04/2009 H1N1 (does not express PB1-F2[44,45]), 13 virus proteins for A/Wyoming/03/2003 H3N2 and 13 virus proteins for A/Vietnam/1203/2004 H5N1 were cloned into a previously described pcDNA4/TO backbone vector[47]. IAV proteins were cloned with either an N-terminal 2X-Strep tag (PB1-F2, NA, M1, M2, NS1, NEP), C-terminal 2X-Strep tag (PB2, PB1, N40, PA, PA-X, NP) or internal 2X-Strep tag (HA). The location for 2X-Strep tag insertion was informed by previously published studies. The 2X-Strep tag was inserted internally into the HA sequence at an insertion permissive site as previously described[115]. The 2X-Strep tag was cloned at the C-terminus of PB1, PB2, N40, PA-X and NP based on successful published functional studies conducted with these proteins tagged at the same position[116,117]. For all other constructs, the 2X-Strep tag was cloned at the N-terminus, as this site was previously used to characterize the M1, M2, NS1 and NEP proteins[113,118,119], and N-terminal fusions are often used to generate recombinant NA[120]. DNA and amino acid sequences for all 2X-Strep-tagged IAV proteins, and 2X-Strep-tagged eGFP and empty vector control proteins, are reported in Supplementary Data 1.

Tagged gene sequences of all IAV proteins were first cloned from the pcDNA4/TO vector into a pLVX-TetOne-Puro doxycycline-inducible backbone vector (Takara, 631847) via Gibson Assembly. Gene inserts derived from PCR amplifications of pcDNA4/TO clones were designed with 15-30 base pairs of overlap with the backbone vector. Seven IAV proteins (PB1, PB1-F2, N40, NA, NS1, NEP, HA) had insufficient expression for AP-MS by this method. To improve protein expression, for these seven IAV proteins from all three strains, gene blocks of tagged constructs were instead codon-optimized using an online codon-optimization tool (Integrated DNA Technologies [IDT]) and synthesized (IDT), and subsequently cloned via Gibson Assembly into the pLVX-TetOne-Puro backbone vector. Gibson Assembly was performed as previously described[121]. Briefly, a 5X ISO Buffer was prepared with 3 mL 1 M Tris-HCl pH 7.5, 150 μL 2 M MgCl₂, 240 μL 100 mM dNTP mix (25 mM each of dGTP, dCTP, dATP, dTTP), 300 μL 1 M DTT, 1.5 g PEG-8000, 600 μL 50 mM NAD 3x (NEB, 9007 S), and dH₂O to 6 mL final volume. 5X ISO Buffer was stored at −20 °C in 320 μL aliquots. A Gibson Assembly master mix was prepared by combining 320 μL of 5X ISO Buffer with 0.64 μL 10 U/μL T5 Exonuclease (NEB, M0363S), 20 μL 2 U/μL Phusion Polymerase (NEB, M0530S), 160 μL 40 U/μL Taq DNA ligase (NEB, M0208L), and water to 1.2 mL final volume. Gibson Assembly mastermix was stored at -20 °C in 15 μL aliquots. The pLVX-TetOne-Puro backbone was linearized with restriction enzymes BamHI-HF (NEB, R3136S) and EcoRI-HF (NEB, R3101S) in accordance with the manufacturer's recommendations. Gibson Assembly reactions were then performed by combining 20 ng of linearized backbone with the insert gene of interest in a 1:2 molar ratio in 15 μL of Gibson master mix plus water to a final volume of 20 μL. Reaction mixtures were then incubated for 30 minutes at 50 °C.

pLVX-TetOne-Puro PA-X-encoding constructs were additionally cloned to include a D108A point mutation in the catalytic RNA endonuclease domain of PA-X. Catalytic IAV PA-X caused cell toxicity; therefore, we cloned a D108A substitution previously shown to inactivate endonuclease activity[122,123]. Briefly, D108A mutagenesis was performed by QuikChange site-directed mutagenesis (Agilent, 200518) on pLVX-TetOne-Puro PA-X constructs following manufacturer's protocol adapted with Velocity enzyme (BioLine, BIO-21099) under the following conditions in a Bio-Rad C1000 Touch Thermal Cycler: 98 °C for 30 seconds - 2 minutes, 18 cycles of 98 °C for 30 seconds followed by 55 °C for 1 minute and 72 °C for 5-10 minutes, and final extension at 72 °C for 3 minutes. pLVX pH1N1 PA-X D108A 2X-Strep and pLVX H5N1 PA-X D108A 2X-Strep were generated by QuikChange mutagenesis alone. H3N2 PA-X D108A 2X-Strep was subjected

to mutagenesis as described above, amplified by PCR with Phusion enzyme (NEB, M0530L), and cloned into empty pLVX-TetOne-Puro vector by InFusion cloning (Takara, 638911) following manufacturer recommendations.

**Stable IAV protein-expressing cell line generation and culture.** A549 cells (ATCC, CCL-185) were cultured in T175 flasks (Fisher, 12-556-011) at 37 °C and 5% CO2 in DMEM with L-glutamine without sodium pyruvate (Fisher, MT 10-017-CV), 10% FBS (Life Technologies, A3160502) and 1X Penicillin/Streptomycin (Pen/Strep) (Fisher, MT 30-002-CI). NHBE cells (Lonza, CC-2541) were cultured in collagen I-coated T175 flasks (Fisher, 356487) at 37 °C and 5% CO2 in Bronchial Epithelial Basal Medium (BEBM) (Lonza, CC-3171) with nine supplemental singlequots from the Bronchial Epithelial Cell Growth Medium (BEGM) kit (Lonza, CC-4175). THP-1 cells (ATCC, TIB-202) were cultured in T175 flasks at 37 °C and 5% CO2 in RPMI-1640 with L-glutamine (Fisher, MT10040CV) supplemented with 10% FBS, 10 mM HEPES (Fisher, SH3023701), 1 mM sodium pyruvate (Fisher, MT 25-000-CI) and 1X Pen/Strep.

For transduction of A549 and NHBE, cells were seeded in appropriate growth media at $5 \times 10^5$ cells per T75 flask (A549) or approximately 2 million cells per collagen I-coated T175 flask (NHBE), transduced with 250–500 μL lentivirus containing the IAV transgene of interest, and returned to incubate at 37 °C for 48 hours. Media was subsequently removed and replaced with appropriate cell growth media supplemented with 1 μg/mL puromycin (A549) or 0.5 μg/mL puromycin (NHBE) for transgene selection. Cells were expanded in selection media as polyclonal pools for four days (A549) or 48 hours (NHBE), to nearly 100% confluence. Cells were then split 1:6 and seeded in six replicates in selection media, equating to about 2 million cells per 15 cm dish (A549) (Fisher, 430599) or collagen I-coated 15 cm dish (NHBE) (Fisher, 08-774-9), and allowed to incubate for further expansion and transgene expression. Transgene expression was induced at three days (A549) and five days (NHBE) after seeding cells into 15 cm format.

For transduction of THP-1, cells were seeded in 2 mL appropriate growth media at 1 million cells per well in a 6-well plate (Fisher, 08-772-1B). Cells were transfected in 6-well plate format with 25 μL lentivirus containing the IAV transgene of interest and returned to incubate at 37 °C for 48 hours. Cells containing the transgene were selected by incubation with growth media supplemented with 0.75 μg/mL puromycin for 72 hours. For subsequent monoclonal selection, cells were serially diluted to 150 cells/mL in growth media supplemented with 0.25 μg/mL puromycin, diluted again 1:40 in selection media and plated into 96-well flat-bottom plates (Fisher, 08-772-2 C). Cells were incubated at 37 °C for 3-4 weeks in selection media to allow single cell colony outgrowth. 12 colonies per transgene were selected, expanded for roughly 12 days in selection media into 24-well plates, and screened for inducible, sufficient transgene expression by doxycycline treatment (below) followed by immunoblot. Four successful monoclonal isolates per transgene were expanded in selection media into T175 flasks to a density of $1 \times 10^6$ cells/mL in a final volume of 100 mL. Following monoclonal expansion, cells were differentiated into a macrophage-like state with phorbol 12-myristate 13-acetate (PMA) (Fisher, BP685-1). Briefly, 25 million THP-1 cells from each of the four monoclonal pools were plated in growth media supplemented with 0.25 μg/mL puromycin and 30 nM PMA in four 15 cm dishes, two dishes per replicate. THP-1 cells were PMA-differentiated for 48 hours before transgene expression was induced. Each monoclonal isolate serves as a replicate for THP-1.

To induce transgene (IAV protein) expression in A549, NHBE and THP-1, cells were treated with doxycycline (Fisher, AAJ6057914) at final concentration 2 μg/mL for a total of 24 hours. 12 hours after doxycycline treatment, one set of replicates was treated with universal type I interferon (PBL Assay Science, 11200-2) at final concentration

1000 U/mL for 12 hours to stimulate an antiviral-like state, and one set of replicates remained untreated. There were few discernible differences in observed PPIs between treated and untreated replicate sets, therefore replicate sets were combined totaling six biological replicates (A549 and NHBE) or eight biological replicates (THP-1) to increase statistical power. To achieve sufficiently high protein levels of PB1-F2 in all cell types, PB1-F2-expressing cells were treated with proteasome inhibitor MG-132 (Sigma-Aldrich, 474790) at final concentration 5 μM at 12 hours after doxycycline treatment for 12 hours before harvest and affinity purification.

**PPI sample harvest and affinity purification.** To harvest 2X-Strep-tagged IAV protein- and control-expressing A549 and NHBE cells, cells were washed in 10 mL 1X phosphate buffered saline (PBS) (Fisher, MT21031CV) and detached from plates by cell scraper (Fisher, 50-809-263) in 10 mL 1X PBS followed by a 4 mL wash for a 14 mL final cell suspension per replicate. THP-1 cells were washed in 10 mL 1X PBS and detached by cell scraper in 10 mL 1X PBS, and two dishes per replicate were combined. Each dish was then washed with an additional 5 mL per plate for a final combined 30 mL cell suspension per replicate. Cells were pelleted at 2000 rpm, 4 °C for 5 minutes, supernatant was aspirated, and pellets were resuspended in 1 mL cold lysis buffer (Immunoprecipitation (IP) buffer pH 7.4 at 4 °C (50 mM Tris-HCl pH 7.5, 150 mM NaCl, 1 mM EDTA (Fisher, MT-46034CI)) supplemented with 0.5% Nonidet P40 substitute (NP40) (United States Biological, 9036-19-5), cOmplete mini EDTA-free protease inhibitor (Roche, 11836153001) and PhosSTOP phosphatase inhibitor (Roche, 04906837001)). Samples were transferred to 1.5 mL epitubes (Fisher, 05-408-129) and rotated at 4 °C for 30 minutes. Samples were subsequently frozen at −80 °C for a minimum of 30 minutes, or until affinity purification.

Affinity purification was performed against the 2X-Strep tag with 50% suspension Strep-Tactin Sepharose beads (IBA, 2-1201-010). 20 μL bead volume (40 μL 50% slurry) per sample was washed in IP buffer pH 7.4 at 4 °C, pelleted at 1000 rpm for 5 minutes and resuspended in 640 μL cold IP buffer per sample (total 660 μL bead suspension). 660 μL bead suspension was then transferred to one 2 mL dolphin tube per sample (VWR, 53550-148). During this time, samples were thawed at room temperature for 20-30 minutes, and clarified by centrifugation at $3500 \times g$, 4 °C for 20 minutes to pellet debris. 50 μl lysate (input) was reserved for immunoblotting. 950 μL remaining lysate per sample was transferred to the corresponding 2 mL dolphin tube containing Strep-Tactin Sepharose beads and incubated for 4 hours at 4 °C with rotation. Beads were subsequently pelleted at 2000 rpm, 4 °C for 4 minutes, and washed twice in 1 mL cold wash buffer (IP buffer pH 7.4 at 4 °C with 0.05% NP40) and twice in 1 mL cold IP buffer (no NP40) by inverting 15 times and pelleting again 2000 rpm, 4 °C for 4 minutes. After the final wash, beads were resuspended in 450 μL cold IP buffer, transferred to lo-bind 0.6 mL epitubes (Axygen, MCT-060-L-C) with wide-orifice tips (Rainin, 17007099) and pelleted at 2000 rpm, 4 °C for 4 minutes. Supernatant was aspirated by 1 mL syringe (BD Biosciences, 309628) and 27-G needle (BD Biosciences, 309659), and beads were immediately processed for on-bead digestion.

**Immunoblotting.** To verify transgene expression in THP-1 monoclonal isolates, 500 μL suspensions of doxycycline-induced cells from a 24-well plate were transferred to 1.5 mL epitubes and pelleted at 8000 rpm for 2 minutes. Supernatant was removed, and cells were washed with 500 μL 1X PBS, pelleted again and resuspended in 100 μL 2.5X reducing sample buffer (31.2 mM Tris-HCl pH 6.8, 10% glycerol, 1% SDS, 0.83% beta-mercaptoethanol, 0.0126% bromophenol blue). Cell samples were vortexed, boiled at 98 °C for 30 minutes, vortexed again and cooled to room temperature before storage at −20 °C. Verification of transgene expression in A549 and NHBE cells was done at the time of affinity purification. To prepare affinity purification samples for

immunoblot, 50 μL input was combined with 50 μL 2.5X reducing sample buffer, vortexed, boiled at 98 °C for 30 minutes, vortexed again and cooled to room temperature before storage at −20 °C.

For immunoblotting, samples were thawed at room temperature, and 10 μL was loaded into each well of a 26-well 4–20% Criterion™ TGX™ Gel (Bio-Rad, 567-1094). Gels were run at 90 volts for 30 minutes followed by 150 volts for 50 minutes. The samples were then transferred at 0.25 amps for 1 hour to a PVDF Membrane (Bio-Rad, 1620177). Following protein transfer, membranes were blocked in 4% milk in PBST for 1 hour at room temperature, and then incubated with 1:1000 mouse anti-STREP (Qiagen, 34850) or 1:5000 mouse anti-GAPDH (GAPDH-71.1) (Sigma, G8795) in blocking solution overnight at 4 °C. Membranes were washed three times in PBST for 5 minutes each, and then incubated with 1:5000 goat anti-mouse IgG-HRP antibody (Bio-Rad, 170-6516) for 1 hour at room temperature. Membranes were washed three times and stained with Pierce ECL Western Blotting Substrate (ThermoFisher, 32106). Exposures of the blots were taken with autoradiography film (Thomas Scientific, XC59X), and developed with a medical film processor (Konica Minolta Medical & Graphic, SRX-101A). Film was scanned at 300 pixels/inch (Epson Perfection V550 Photo) and stored as 8 bit grayscale TIFF files. Immunoblotting was performed separately for each IAV protein in each cell type as samples were generated; however, homologous IAV proteins from all three strains within one cell type were processed concurrently. TIFF files from all IAV proteins and all cell types were collated for publication (Supplementary Fig. 3), with the following modifications: brightness via the levels adjustment tool in Photoshop applied equally across each independent film scan to normalize background across different film, and image scaling and perspective editing via the perspective warp tool in Photoshop to horizontally align samples.

**On-bead digestion and peptide desalting.** Bead-bound proteins were reduced and alkylated by incubation in one bead volume equivalent of reduction/alkylation buffer (2 M urea, 50 mM Tris pH 8.0, 1 mM Dithiothreitol [DTT] (Sigma, D5545), 3 mM iodoacetamide (Sigma, I1149) in HPLC-grade water (Fisher Chemical, W7-4)) for 45 minutes in the dark with gentle agitation to ensure bead suspension. Iodoacetamide was quenched with an additional 3 mM DTT. Bead-bound proteins were then digested by incubation with 750 ng sequencing-grade trypsin (Promega, V5111) per 10 μL bead volume, and incubated overnight at 37 °C. Resulting in-solution peptides were extracted from beads by gel-loading tips (Fisher, 02-707-81) into a fresh 0.6 mL lo-bind epitube for each sample and acidified by addition of HPLC-grade formic acid (FA) (Fisher Chemical, A117-50) to final concentration 1%.

Acidified peptides were desalted for MS analysis using HPLC-grade reagents and OMIX C18 10 μL tips (Agilent Technologies, A5700310K) according to manufacturer's protocol. Briefly, OMIX tips were conditioned with 50% acetonitrile, 0.1% FA and equilibrated with two washes of 0.1% FA. Peptides were bound to C18 zip-tip by repeated rinsing. Polymer-bound peptides were washed three times with 0.1% FA and eluted in 50% acetonitrile, 0.1% FA. A second elution in 90% acetonitrile, 0.1% FA was performed to increase peptide recovery. Peptides were dried by speedvac vacuum centrifugation (CentriVap Concentrator with CentriVap Cold Trap, Labconco) and stored at −20 °C until MS analysis.

**PPI MS data acquisition and analysis.** Digested, desalted and dried peptides were dissolved in 12 μL 2% acetonitrile, 0.1% FA. 2 μL of each sample were injected in technical singlet for LC-MS/MS analysis onto an Easy-nLC 1000 (Thermo Fisher Scientific) interfaced with an Orbitrap Elite Hybrid Mass Spectrometer (Thermo Fisher Scientific). Briefly, peptides were separated on a 75 μm x 25 cm fused silica IntegraFrit capillary packed with 1.9 μm Reprosil-Pur C18 AQ reversed-phase resin (Dr. Maisch GMBH, r119.aq) over a 120-minute gradient at a flow rate of 300 nL/minute as described in Supplementary Data 6. Buffer A

consisted of 0.1% FA in water, and buffer B was 0.1% FA in acetonitrile. For each cycle, one full MS scan in the Orbitrap (150–1500 m/z, at 120,000 resolution with an AGC target of $1 \times 10^6$ and maximum injection time of 100 milliseconds) was followed by 20 data-dependent MS/MS scans acquired in the linear ion trap (AGC target $3 \times 10^4$, maximum injection time of 50 milliseconds, fragmented by normalized collision energy at 35%). Target ions already acquired in MS/MS scans were dynamically excluded for 20 seconds (tolerance of 10 ppm). Detailed MS acquisition parameters are reported in Supplementary Data 6.

Raw MS files from IAV proteins from all strains and control protein samples were grouped separately by cell line and searched simultaneously within each group using MaxQuant (version 1.6.2.10)[46]. 6 replicates were searched for each IAV protein and control protein in A549 and NHBE MaxQuant groups; 7–8 replicates were searched for each IAV protein and control protein in THP-1 MaxQuant group. MS/MS spectra were searched against the human proteome (SwissProt human canonical sequences, downloaded 09 October 2018), IAV protein sequences and the eGFP sequence. Trypsin (KR | P) was selected to allow up to two missed cleavages. Variable modifications were assigned for: methionine oxidation and N-terminal protein acetylation. One static modification was assigned for carbamidomethyl cysteine. Label free quantitation (LFQ) was enabled. All other MaxQuant settings were left at the default.

MaxQuant-analyzed data were then scored using the MiST algorithm[47] that assigns quantitative interaction confidence scores based on specificity, abundance and reproducibility, following previous guidelines[124] using spectral counts as the quantifying feature. To enable robust scoring, we excluded samples with low spectral counts and low or no IAV protein identification, and those with less than two replicates. Following these quality control filtering steps, 590 samples across 14 baits (12 IAV proteins and 2 control proteins) remained for analysis. We ran the MiST algorithm applying a weight set of specificity S = 50%, reproducibility R = 45% and abundance A = 5%. To identify high-confidence PPIs, we applied a set of stringent scoring criteria: (1) MiST score > 0.6; (2) the interaction is absent in eGFP and empty-vector control samples; and (3) at least 4/6 replicates have a spectral count > 0. IAV M2 protein in A549 cells and IAV NP protein in NHBE cells retained a disproportionately large number of interactions, therefore we applied more stringent scoring criteria to these two specific samples: (1) MiST score > 0.75; (2) the interaction is absent in eGFP and empty-vector control samples; (3) at least 5/6 replicates have a spectral count > 0; and (4) average spectral count > 3. Interactions that fall above these cutoffs represent the final high-confidence PPI list, and contain a total of 126 interactions in A549 (top 4% of interactions), 130 interactions in NHBE (top 9% of interactions), and 76 interactions in THP-1 (top 5% of interactions) (Supplementary Data 1). This list was used for further bioinformatic analyses and validation.

## Validation of IAV-human PPIs M2-ATP6V1A and NEP-AHNAK in IAV-infected cells

**Viruses.** IAV isolate A/Vietnam/1203/2004 H5N1 HALo was rescued by reverse genetics from available sequences and engineered to contain a deletion in the HA polybasic cleavage site[125], allowing its use in Biosafety Level 2+ facilities. All IAV viruses were propagated in embryonated chicken eggs (Charles River Laboratories) following methods previously described[126]. Virus stocks were titrated in Madin-Darby canine kidney (MDCK) cells (ATCC, CCL-34) by plaque assay. All IAV infections were performed according to institutional Biosafety Level 2+ biosafety procedures at the J. David Gladstone Institutes.

**Cell infections and harvest for reciprocal IPs, flow cytometry and immunofluorescence (IF).** For reciprocal IPs, A549 and NHBE cells were cultured as described above (IAV-Human PPI AP-MS Methods). One 15 cm dish was seeded for each IP from both mock and H5N1 infection conditions in A549 cells (ATP6V1A or IgG baits) or NHBE cells

(AHNAK or IgG1 baits). A549 cells were seeded at $5 \times 10^6$ million cells per 15 cm tissue culture dish (Thermo Fisher, 130183), cultured in DMEM media with 10% FBS and 1X Pen/Strep, and expanded to roughly $2 \times 10^7$ cells at time of infection. NHBE cells were seeded at $2 \times 10^6$ cells per collagen I-coated 15 cm dish (Corning, 354551), cultured in BEBM media with nine BEGM supplemental singlequots, and expanded to $7 \times 10^6$ cells at time of infection. For IAV infections, cell growth media was removed, and A549 or NHBE cells were infected with A/Vietnam/1203/2004 H5N1 HALo at MOI 0.5 in 10 mL 0.5% Bovine Serum Albumin (BSA) (GoldBio, A-420-100) in 1X PBS per plate, or mock-infected with 10 mL 0.5% BSA in 1X PBS per plate. Cells were incubated for 1 hour at 37 °C and 5% CO2 and rocked by hand every 10 minutes, after which inoculum was removed and replaced with 25 mL DMEM with 0.1% FBS, 0.3% BSA and 0.5 µg/mL TPCK-treated trypsin (Sigma-Aldrich, T8802) per plate (A549 cells) or 27 mL BEBM growth media with 0.25 µg/mL TPCK-treated trypsin per plate (NHBE cells). Cells were returned to incubate at 37 °C and harvested at 24 hours post-infection. Briefly, media was removed, and cells were detached from plates by cell scraper in 10 mL cold 1X PBS and collected in 15 mL Falcon tubes (Fisher, 14-959-53 A). Cells were pelleted at 1200 rpm for 5 minutes at 4 °C (A549 cells), or $400 \times g$ for 3 minutes at 4 °C (NHBE cells). The supernatant was removed, and pellets were stored on ice. 10 mL cold 1X PBS was added to the original 15 cm plates for a second cell scraping, collected and used to resuspend and wash existing cell pellets. 100 µL aliquots were removed and combined with 100 µL 2% formaldehyde (Sigma, F8775-500ML) in 1X PBS in a 96-well U-bottom plate (Fisher 08-772-17) for flow cytometry. Cells were pelleted again as described above, washed a second time in 10 mL cold 1X PBS, and pelleted again, after which supernatant was removed. Cell pellets were frozen at −80 °C overnight, and subsequently thawed on ice in 1 mL cold lysis buffer (recipe described above in IAV-Human PPI AP-MS Methods). Samples were transferred to lo-bind 1.5 mL epitubes (Fisher, 022431081) and rotated end-over-end at 4 °C for 30 minutes. Samples were subsequently frozen at −80 °C overnight until reciprocal IP.

For IF, NHBE cells were seeded at $5 \times 10^4$ cells per well in collagen I-coated 24-well plates (Corning, 354408) and cultured as described above in BEBM media with nine BEGM supplemental singlequots. For IAV and mock infections, cell growth media was removed and replaced with 400 µL per well A/Vietnam/1203/2004 H5N1 HALo at MOI 0.5 in 0.5% BSA in 1X PBS, or 400 µL per well 0.5% BSA in 1X PBS. Cells were incubated for 1 hour at 37 °C and 5% CO2 and rocked by hand every 10 minutes. After virus adsorption, inoculum was removed and replaced with 400 µL per well BEBM growth media with 0.25 µg/mL TPCK-treated trypsin. At 24 hours post-infection, media was removed, and cells were washed in 300 µL 1X PBS and fixed to the collagen I-coated 24-well plate in 300 µL 4% formaldehyde in 1X PBS for 15 minutes at room temperature. Fixative was removed, cells were washed again in 300 µL 1X PBS and then permeabilized with 300 µL 0.1% TritonX (Sigma, T8787) in 1X PBS for 15 minutes at room temperature. TritonX solution was removed, and the fixed, permeabilized cells were stored in 1 mL 1X PBS at 4 °C, with the plate sealed by parafilm and wrapped in aluminum foil, until IF.

**Reciprocal IPs against ATP6V1A and AHNAK.** Cell lysates were thawed on ice and clarified by centrifugation at $3500 \times g$ for 20 minutes at 4 °C. 50 µL of clarified lysate was reserved as input aliquots for each sample and subjected to Bradford assay for total protein quantification and targeted MS analysis for endogenous protein and IAV protein quantification. Total protein in each sample was quantified by Bradford Reagent (Sigma, B6916) following manufacturer instructions on a SpectraMax iD3 microplate reader (Molecular Devices) with acquisition at 595 nm using manufacturer software SoftMax Pro (version 7.1) (Molecular Devices). For targeted MS analysis, roughly 25–50 µg protein from the input samples was reduced in 4 mM tris(2-carboxyethyl)phosphine (TCEP) (Thermo Fisher, 20491) for

20 minutes at room temperature, alkylated in 10 mM iodoacetamide for 20 minutes at room temperature in the dark, and quenched in 10 mM DTT at room temperature for 5 minutes. Reduced and alkylated input samples were then subjected to methanol chloroform precipitation. Briefly, 1 part sample was combined and vortexed sequentially with 4 parts methanol, 1 part chloroform and 3 parts water for phase separation. Samples were spun in a bench-top centrifuge (Centrifuge 5424 R, Eppendorf) for 2 minutes at $14,000 \times g$, after which the upper phase was discarded. 4 parts methanol was added and vortexed with the interphase and lower phase, samples were centrifuged for 8 minutes at $14,000 \times g$, and the supernatant was discarded. The protein pellet was washed three times in 1 mL 80% ice cold acetone with centrifugation for 8 minutes at $14,000 \times g$ after each wash. Precipitated proteins were air dried, resuspended in 8 M urea, 50 mM ammonium bicarbonate in HPLC-grade water, diluted 5-fold in 50 mM ammonium bicarbonate in HPLC-grade water, and digested with sequencing-grade trypsin at a 1:100 (enzyme:protein w-w) ratio overnight at 37 °C in a thermomixer (Thermomixer C, Eppendorf) at 500 rpm. After digestion, peptides were acidified by addition of HPLC-grade FA to final concentration 1% before desalting.

For endogenous IPs, 950 µL-1 mL of clarified lysate (roughly 0.8–1.6 mg total protein) from A549 cells was combined with 1:100 (antibody:lysate v-v) anti-ATP6V1A rabbit monoclonal (EPR19270) antibody (Abcam, ab199326) or 1:200 (antibody:lysate v-v) rabbit IgG control antibody (Proteintech, 501003118) in a 1.5 mL lo-bind epitube. 950 µL-1 mL of clarified lysate (roughly 0.3-0.5 mg total protein) from NHBE cells was combined with 1:100 (antibody:lysate v-v) anti-AHNAK mouse monoclonal (EM-09) antibody (Thermo Fisher, MA1-10050) or 1:200 (antibody:lysate v-v) mouse IgG1 isotype control (MOPC-21) antibody (Thermo Fisher, MA1-10407) in a 1.5 mL lo-bind epitube. Lysate-primary antibody mixtures were incubated for 16 hours at 4 °C with end-over-end rotation. Purification was performed with 50 µL per sample Pierce Protein A + G magnetic beads (Thermo Fisher, 88802). Beads were washed four times with 1 mL cold lysis buffer (recipe described above in IAV-Human PPI AP-MS Methods) by repeated bead binding to a DynaMag-2 magnetic tube rack (Thermo Fisher, 12321D) for 1-2 minutes, removing supernatant, and resuspending beads thoroughly using wide orifice tips. 50 µL washed beads were incubated with the lysate-primary antibody mixture for 2 hours at 4 °C with end-over-end rotation. Samples were then briefly spun down and washed two times in 1 mL cold wash buffer (IP buffer pH 7.4 at 4 °C with 0.05% NP40) and two times in 1 mL cold IP buffer (no NP40) by repeated bead binding to a DynaMag-2 magnetic tube rack described above. On-bead reduction, alkylation and digestion with sequencing-grade trypsin was performed as described above (IAV-Human PPI AP-MS Methods). Resulting digested, in-solution peptides were extracted from the beads and acidified by addition of HPLC-grade FA to final concentration 1% before desalting, as described above (IAV-Human PPI AP-MS Methods).

For all input and reciprocal IP samples, peptides were desalted by OMIX C18 100 µL tips (Agilent, A57003100K) and dried down by vacuum centrifugation as described above (IAV-Human PPI AP-MS Methods). Digested, desalted and dried peptides were resuspended in 10 µL (input samples) or 12 µL (IP samples) of 3% acetonitrile, 2% FA for targeted MS acquisition.

**Targeted MS data acquisition and analysis.** Targeted MS relied on first queuing data-dependent acquisition (DDA) runs on all reciprocal IP samples to select peptides for parallel reaction monitoring (PRM) acquisition and analysis. PRM was subsequently performed for all input and reciprocal IP samples. DDA data and PRM data were acquired on an Orbitrap Fusion Lumos Tribrid (Thermo Fisher Scientific) interfaced with an Easy-nLC 1200 (Thermo Fisher Scientific). Detailed PRM LC and MS acquisition parameters are reported in Supplementary Data 6.

For DDA-based acquisition, 1 µL of each reciprocal IP sample (A549 cells) or 2 µL of each reciprocal IP sample (NHBE cells) was

separated on a PepSep C18 column (15 cm × 150 µm, 1.9 µm particle size) (Bruker, 1893471) over the course of a 60-minute data acquisition. Buffer A consisted of 0.1% FA in water, and buffer B was 0.1% FA in 80% acetonitrile. Spectra were acquired continuously in a data-dependent manner. One full scan in the Orbitrap (at 240,000 resolution in profile mode with an AGC target of $1 \times 10^6$ and maximum injection time of 50 milliseconds) was followed by as many MS/MS scans as could be acquired on the N most abundant ions in 1 second in the ion trap (rapid scan type, HCD collision energy of 32%, AGC target of $3 \times 10^4$, maximum injection time of 18 milliseconds). Singly and unassigned charge states were rejected. Dynamic exclusion was enabled with a repeat count of 2, an exclusion duration of 20 seconds, and an exclusion mass width of ±10 ppm. Raw files were searched using MaxQuant (version 1.6.3.3)[46] in two search groups: (1) ATP6V1A and IgG baits from mock-infected and H5N1-infected A549 cells; and (2) AHNAK and IgG1 baits from mock-infected and H5N1-infected NHBE cells. MS/MS spectra were searched against the human proteome (SwissProt human canonical sequences, downloaded 09 October 2018), IAV protein sequences and the eGFP sequence. Trypsin (KR | P) was selected to allow up to two missed cleavages. Variable modifications were assigned for: methionine oxidation and N-terminal protein acetylation. One static modification was assigned for carbamidomethyl cysteine. Label free quantitation (LFQ) was enabled. All other MaxQuant settings were left at default. The msms.txt results from the DDA-based MaxQuant searches were then imported into Skyline[127] (version 21.2.0.536 dbaf6ccd2) to select peptides for PRM acquisition and analysis. In the ATP6V1A purification experiment, 8 peptides for ATP6V1A and 2 peptides for M2 were employed. In the AHNAK purification experiment, 6 peptides from AHNAK and 3 peptides from NEP were utilized.

For PRM-based acquisition, 1 µL of each ATP6V1A or IgG reciprocal IP sample (A549 cells), 2 µL of each AHNAK or IgG1 reciprocal IP sample (NHBE cells), or 1 µL of each input sample (NHBE and A549 cells) were separated on the same PepSep column described above using a similar gradient to the bait-matched DDA runs over a 60-minute data acquisition (Supplementary Data 6). The acquisition was performed in parallel reaction monitoring mode, using a time-window scheduling of 4 minutes per peptide, 0.9 m/z isolation window, 60,000 resolution (400 m/z), HCD peptide fragmentation (normalized collision energy of 33%), and AGC target of $1 \times 10^7$. The injection time was set to "dynamic", with 9 points per peak as the target number of points. Detailed MS acquisition parameters are reported in Supplementary Data 6.

Following PRM acquisition, PRM data was extracted in Skyline[127] (version 21.2.0.536 dbaf6ccd2) with the following settings: MS/MS filtering was set to targeted using Orbitrap as the mass analyzer (60,000 resolution, high selectivity extraction), and MS1 filtering was disabled. A minimum of 6 and a maximum of 30 transitions from the library were allowed. After manual peak integration and removal of interfering transitions, the quantification per fragment was exported. Fragments having m/z < precursors or signal/background ratio less than 5 were removed to ensure robust quantitative accuracy. The transitions per peptides were summed and then the average across all peptides for a specific protein were used for quantification. For fold change calculations, the data was log2-transformed and the ratio of the IgG H5N1-infected A549 cell sample, or IgG1 H5N1-infected NHBE cell sample, was used to derive M2 enrichment from ATP6V1A-bait samples, or NEP enrichment from AHNAK-bait samples, respectively (Supplementary Data 1).

**IAV infection staining and flow cytometry.** Whole, intact A549 or NHBE cells harvested and reserved from the 15 cm reciprocal IP plates were immunostained for IAV NP protein and quantified by flow cytometry to determine percent cells infected with IAV. To remove and exchange buffers between incubations and washes, cells were pelleted in 96-well U-bottom plates at $800 \times g$ for 3 minutes. Fixed cells were pelleted and incubated in 100 µL block and permeabilization buffer (1% BSA, 0.1% saponin (Sigma, 47036-50G-F) in 1X PBS) for 30 minutes at room temperature. Cells were pelleted and incubated with 100 µL 1:1000 mouse anti-Influenza A nucleoprotein [HT103] antibody (Kerafast, EMS010) in block and permeabilization buffer for 1 hour at room temperature. Cells were pelleted, washed once with 200 µL wash buffer (1% BSA in 1X PBS) and incubated in 100 µL 1:1000 goat anti-mouse IgG (H + L) Alexa Fluor Plus 488 (Fisher, A32723) in block and permeabilization buffer for 1 hour at room temperature in the dark. Cells were washed twice with 200 µL wash buffer and fixed in 150 µL 1% formaldehyde in 1X PBS. Samples were run in 96-well format on an Attune NxT flow cytometer (Thermo Fisher) with accompanying Attune NxT Software (version 3.2.0). Voltages were set at: (1) forward scatter 80 and side scatter voltage at 275 (R1, A549 singlet cells); and (2) forward scatter 120 and side scatter 280 (NHBE cells). In both experiments, Alexa Fluor Plus 488 signal was quantified by blue laser 1 at voltage 205 (percent NP + ). 100 µL of cells were acquisitioned and all events recorded at 1000 µL/min. Final cell gating and quantification of %NP+ cells (percent IAV infectivity) was performed with FlowJo software (version 10.7.1). %NP+ cells (percent IAV infection) are reported in technical singlet (Supplementary Fig. 5B,D).

**IF staining against AHNAK and IAV NEP and microscopy.** Fixed, permeabilized cells in 24-well plate format were blocked in 300 µL 1% BSA in 1X PBS for 30 minutes at room temperature with gentle rocking, and then incubated in 300 µL 1:500 anti-AHNAK mouse monoclonal (EM-09) (Thermo Fisher, MA1-10050) and 1:500 anti-influenza A NS2 (NEP) rabbit polyclonal (Thermo Fisher, PA5-32234) primary antibodies diluted in 1% BSA in 1X PBS at 4 °C overnight with gentle rocking. The next day, cells were washed three times in 1X PBS for 5 minutes each wash with gentle rocking, and then incubated in 300 µL 1:1000 goat anti-mouse IgG (H + L) Alexa Fluor 555 (Thermo Fisher, A32727) and 1:1000 goat anti-rabbit IgG (H + L) Alexa Fluor 647 (Thermo Fisher, A32733) secondary antibodies diluted in 1% BSA in 1X PBS for 1 hour at room temperature, with the plate wrapped in aluminum foil to protect against light exposure and gentle rocking. Cells were then washed three times as previously described with the plate wrapped in aluminum foil. Cells were finally stained in 300 µL 1:2000 Hoechst 33342 nuclear stain (Invitrogen, H3570) for 30 minutes at room temperature, with the plate wrapped in aluminum foil and gentle rocking. Cells were washed three times as previously described with the plate wrapped in aluminum foil, and suspended in 300 µL 1X PBS. The stained 24-well plate was sealed by parafilm and stored at 4 °C wrapped in aluminum foil until microscopy.

For microscopy, cells were imaged directly in multi-well format in the 24-well collagen I-coated plate in which they were cultured, infected, fixed and stained. Images were acquired using MetaXpress 6 software on an ImageXpress Micro Confocal High-Content Screening System (Molecular Devices) using the included 10X Plan Apo 0.45 NA objective and 60 µm pinhole spinning disk. Image overlays and montages were formatted with Molecular Devices MetaXpress Imaging and Analysis software suite (version 6.7.2.290). Additional formatting of scale bars and figure panels was performed in Adobe Photoshop (version 23.0.2) and Illustrator (version 26.0.1), respectively.

### Global abundance and phosphorylation profiling methods

**Viruses.** IAV isolates A/California/04/2009 H1N1 and A/Wyoming/03/2003 H3N2 were obtained through BEI Resources (NIAID, NIH). A/Vietnam/1203/2004 H5N1 HALo was generated as described above (Validation of IAV-Human PPIs M2-ATP6V1A and NEP-AHNAK in IAV-Infected Cells Methods). All IAV viruses were propagated in embryonated chicken eggs (Charles River Laboratories) following methods previously described[126]. Virus stocks were titrated in MDCK cells by plaque assay. All infections with live IAV were performed in accordance with institutional Biosafety Level 2+ biosafety procedures at the Icahn

School of Medicine at Mount Sinai and the J. David Gladstone Institutes.

**Cell infections for global proteomic analysis.** NHBE cells were seeded at $1 \times 10^7$ cells per collagen I-coated 15 cm dish and cultured in BEBM media with nine BEGM supplemental singlequots (described above). THP-1 cells were expanded in suspension in T75 flasks at a density of $2 \times 10^5$–$8 \times 10^5$ cells/mL and cultured in RPMI-1640 with L-glutamine supplemented with 10% FBS, gentamicin (Thermo Scientific, 15750060) at final concentration 50 μg/mL and 1X Pen/Strep. For plating, THP-1 cells were pelleted at $500 \times g$ for 5 minutes, resuspended in growth media supplemented with PMA at final concentration 10 ng/mL to induce differentiation, and subsequently seeded at $2 \times 10^7$ cells per 15 cm dish. THP-1 cells were differentiated for 72 hours in PMA media, before media was exchanged with growth media (no PMA) for 24 hours to reduce PMA-activated pro-inflammatory response. For infection, cell growth media was removed, and cells were either mock infected or infected in biological duplicate with A/California/04/2009 H1N1 IAV, A/Wyoming/03/2003 H3N2 IAV or A/Vietnam/1203/2004 H5N1 HALo IAV at MOI 2 in 0.5% BSA in 1X PBS with magnesium and calcium. Cells were incubated in virus inoculum at 37 °C for 1 hour. After adsorption, virus inoculum was aspirated and replaced with cell growth media supplemented with TPCK-trypsin. Cells were returned to incubate at 37 °C before cell harvest and global proteomics sample preparation and processing.

**Global proteomics sample preparation.** IAV-infected cells were harvested and lysed at 3 hours, 6 hours, 12 hours and 18 hours post-infection in biological duplicate with time point-matched mocks for each IAV strain. At the indicated time point, cells were washed with 1X PBS and lysed in 2 mL urea lysis buffer (8 M urea, 100 mM Tris pH 8.0, 150 mM NaCl) supplemented with cOmplete mini EDTA-free protease inhibitor and PhosSTOP phosphatase inhibitor. Cells were harvested in lysis buffer by cell scraper, collected in 15 mL Falcon tubes and incubated on ice for 30 minutes. Samples were subsequently snap-frozen in liquid nitrogen and stored at −80 °C until probe sonication. Samples were thawed on ice and subjected to three rounds of probe sonication (Fisherbrand™ Model 505 Sonic Dismembrator) at 20% amplitude for 20 seconds followed by 10 seconds of rest on ice. Protein concentration was then determined by Bradford assay. Protein from clarified lysate was reduced with 4 mM TCEP for 30 minutes at room temperature, and alkylated with iodoacetamide at final concentration 10 mM for 30 minutes at room temperature in the dark. Iodoacetamide was quenched by addition of final concentration 10 mM DTT and incubation in the dark at room temperature for 30 minutes. For digestion, samples were diluted with 0.1 M ammonium bicarbonate pH 8.0 to a final concentration of 2 M urea. Sequencing-grade trypsin was added at a 1:100 (enzyme:protein w-w) ratio and incubated overnight at 37 °C. Following digestion, 10% trifluoroacetic acid (TFA) was added to acidify each sample to a final pH ~2. Samples were desalted by vacuum manifold (Thermo Fisher Scientific) using Sep Pak tC18 cartridges (Waters, WAT054955) and HPLC-grade reagents. Each cartridge was activated with 1 mL 80% acetonitrile, 0.1% TFA, and equilibrated three times with 1 mL 0.1% TFA. Samples were loaded onto C18 cartridges, and peptide-bound cartridges were washed four times with 1 mL 0.1% TFA. Samples were then eluted four times with 0.5 mL 50% acetonitrile, 0.25% FA to maximize peptide recovery. 10 μg of each sample was reserved for global protein abundance MS data acquisition, and the remainder (at least 1 mg) was allocated to phosphopeptide enrichment. All samples were dried by speedvac vacuum centrifugation (CentriVap Concentrator with CentriVap Cold Trap, Labconco).

**Phosphopeptide enrichment.** For phosphopeptide enrichment, iron nitriloacetic acid (NTA) agarose resin was prepared in-house from 50%

nickel NTA (Ni-NTA) Superflow bead slurry (Qiagen, 30210). 30 μL per sample of 50% Ni-NTA Superflow bead slurry was added to a 2 mL bio-spin column (Bio-Rad, 732-6204). Beads were stripped of nickel ions by four 30-second incubations with 500 μL 100 mM EDTA. Beads were conditioned and loaded with iron by two washes with 500 μL H2O, four 1-minute incubations with 500 μL 100 mM FeCl3, three washes with 500 μL H2O, and one wash with 500 μL 0.5% FA to remove residual iron. Beads were resuspended in 600 μL H2O, and 60 μL was aliquoted into a C18 NEST column (Fisher, NC0484000) that was equilibrated with 150 μL of 80% acetonitrile, 0.1% TFA. 1 mg of digested, dried peptides were resuspended in 75% acetonitrile, 0.15% TFA. Peptides were incubated with the beads for 2 minutes, mixed by pipetting and incubated again for 2 minutes. Beads were washed four times with 200 μL 80% acetonitrile, 0.1% TFA, followed by three washes with 200 μL 0.5% FA. Beads were then incubated twice with 200 μL 500 mM potassium phosphate buffer pH 7 for 15 seconds, and twice with 200 μL 0.5% FA for 15 seconds. Phosphopeptides were eluted twice to maximize recovery with 75 μL 50% acetonitrile, 0.25% FA by centrifugation at 3000 rpm for 30 seconds, and dried by speedvac vacuum centrifugation (CentriVap Concentrator with CentriVap Cold Trap, Labconco).

**Global phosphorylation and abundance MS data acquisition and analysis.** Global AB and PH MS samples were collected on three instruments following instrument-specific LC and MS acquisition parameters (Supplementary Data 6). Samples acquired on an Orbitrap Fusion Tribrid mass spectrometer (Thermo Fisher Scientific) include: (1) NHBE AB data for pH1N1, H3N2 and H5N1; (2) NHBE PH data for pH1N1 and H3N2; and (3) THP-1 AB data for pH1N1, H3N2 and H5N1. Samples acquired on an Orbitrap Elite Hybrid Mass Spectrometer (Thermo Fisher Scientific) include: NHBE PH data for H5N1. Samples acquired on an Orbitrap Fusion Lumos Tribrid mass spectrometer (Thermo Fisher Scientific) include: THP-1 PH data for pH1N1, H3N2 and H5N1.

For samples acquired on the Orbitrap Fusion Tribrid, digested, desalted and dried peptides were resuspended in 10 μL 0.1% TFA (AB samples) or 15 μL of 0.1% TFA (PH samples). 2 μL of each sample were injected in technical duplicate (samples from NHBE cells) or technical singlet (samples from THP-1 cells) on an Easy-nLC 1000 (Thermo Fisher Scientific) interfaced with an Orbitrap Fusion Tribrid mass spectrometer (Thermo Fisher Scientific). Briefly, peptides were separated on a 75 μm × 25 cm fused silica IntegraFrit capillary packed with 1.9 μm Reprosil-Pur C18 AQ reversed-phase resin (Dr. Maisch GMBH, r119.aq) over a 180-minute gradient at a flow rate of 300 nL/minute as described in Supplementary Data 6. Buffer A consisted of 0.1% FA in water, and buffer B was 0.1% FA in acetonitrile. Spectra were continuously acquired in a data-dependent manner. One full scan in the Orbitrap (400-1600 m/z at 120,000 resolution with an AGC target of $2 \times 10^5$ and maximum injection time of 100 milliseconds) was followed by as many MS/MS scans as could be acquired on the most abundant ions in 3 seconds in the dual linear ion trap (HCD collision energy of 30%, AGC target of $1 \times 10^4$, maximum injection time of 35 milliseconds, and isolation window of 1.6 m/z). Singly and unassigned charge states were rejected. Dynamic exclusion was enabled after $n = 1$ time, with an exclusion duration of 40 seconds (tolerance of ±10 ppm). Detailed MS acquisition parameters are reported in Supplementary Data 6.

For samples acquired on an Orbitrap Elite Hybrid Mass Spectrometer, digested, desalted and dried peptides were resuspended in 10 μL 0.1% TFA (AB samples) or 15 μL of 0.1% TFA (PH samples). 2 μL of each sample were injected in technical duplicate on an Easy-nLC 1000 (Thermo Fisher Scientific) interfaced with a Orbitrap Elite Hybrid Mass Spectrometer (Thermo Fisher Scientific). Briefly, peptides were separated on a 75 μm × 25 cm fused silica IntegraFrit capillary packed with 1.9 μm Reprosil-Pur C18 AQ reversed-phase resin (Dr. Maisch GMBH, r119.aq) over a 240-minute gradient at a flow rate of 300 nL/minute as

described in Supplementary Data 6. Buffer A consisted of 0.1% FA in water, and buffer B was 0.1% FA in acetonitrile. Spectra were continuously acquired in a data-dependent manner. For each cycle, one full scan in the Orbitrap (200-2000 m/z, at 120,000 resolution with an AGC target of $1 \times 10^6$ and maximum injection time of 100 milliseconds) was followed by 20 MS/MS scans acquired in the linear ion trap (AGC target of $3 \times 10^4$, maximum injection time of 50 ms, fragmented by normalized collision energy at 35%). Singly and unassigned charge states were rejected. Dynamic exclusion was enabled with a repeat count of 1, an exclusion duration of 20 seconds (tolerance of ±10 ppm). Detailed MS acquisition parameters are reported in Supplementary Data 6.

For samples acquired on the Orbitrap Fusion Lumos Tribrid, digested, desalted and dried peptides were resuspended in 15 µL of 4% FA, 3% acetonitrile. 2 µL of each sample were injected in technical singlet onto an Easy-nLC 1200 (Thermo Fisher Scientific) interfaced via a nanoelectrospray source (Nanospray Flex) with an Orbitrap Fusion Lumos Tribrid mass spectrometer (Thermo Fisher Scientific). Briefly, peptides were separated on a C18 reverse phase column (75 µm × 25 cm packed with 1.9 µm Reprosil-Pur C18 AQ reversed-phase resin) over the course of a 180-minute data acquisition as described in Supplementary Data 6. Buffer A consisted of 0.1% FA in water, and buffer B was 0.1% FA in acetonitrile. Spectra were continuously acquired in a data-dependent manner. One full scan in the Orbitrap (at 120,000 resolution in profile mode with an AGC target of $2 \times 10^5$ and maximum injection time of 100 milliseconds) was followed by as many MS/MS scans as could be acquired on the most abundant ions in 3 seconds in the dual linear ion trap (rapid scan type with an intensity threshold of 5000, HCD collision energy of 30%, AGC target of $1 \times 10^4$, maximum injection time of 35 milliseconds, and isolation width of 1.6 m/z). Singly and unassigned charge states were rejected. Dynamic exclusion was enabled with a repeat count of 1, an exclusion duration of 30 seconds, and an exclusion mass width of ±10 ppm. Detailed MS acquisition parameters are reported in Supplementary Data 6.

Raw MS files from IAV infection time course samples were grouped separately by cell line, enrichment (abundance vs phosphorylation), and instrument, and searched simultaneously within each group using MaxQuant (version 1.6.1.0)[46]. MS/MS spectra were searched against the human proteome (SwissProt human canonical sequences, downloaded 09 October 2018) and IAV protein sequences. Trypsin (KR|P) was selected to allow up to two missed cleavages. Variable modifications were assigned for: N-terminal protein acetylation, N-terminal protein methionine oxidation, and phosphorylation of serine, threonine, and tyrosine (the latter for phosphorylation enrichment samples only). One static modification was assigned for carbamidomethyl cysteine. LFQ was enabled. Match between runs was enabled with a 1.5-minute matching time window and 20-minute alignment window. All other MaxQuant settings were left at the default.

Peptide ion intensities from the output of MaxQuant were summarized to protein intensities using the R Bioconductor package MSstats (version 3.19.4)[69], specifically the function dataProcess, with default settings except that the noise-filtering[128] was turned on by setting featureSubset = "highQuality" and remove_uninformative_feature_outlier = TRUE. For phosphopeptide data, the peptide ion intensities were similarly summarized to a single intensity per unique observed single-peptide combination of phosphorylated sites by relabeling the protein of each feature as the combination of protein name and observed phosphorylated sites. The Bioconductor package artMS (version 1.3.9) (https://doi.org/10.18129/B9.bioc.artMS) was used for this relabeling. The differences in log2-transformed intensity between infected and mock samples were scored using the MSstats function groupComparison, which fits a single linear model for each protein with a single categorical variable for condition. From these models, MSstats reports pairwise differences in means between

conditions as log2 fold change (log2FC) with a p-value based on a t-test assuming equal variance across all conditions, and reports adjusted p-values using the false discovery rate (FDR) estimated by the Benjamini-Hochberg procedure. One single time point per virus was selected for both cell types based on high viral protein abundance in the abundance data: at 18 hours post-infection for pH1N1 and H3N2, and 12 hours post-infection for H5N1. To determine significant changes in protein abundance and phosphorylation, selection criteria included: (1) adjusted p-value < 0.05; and (2) absolute(log2FC) > 1 (Supplementary Data 2).

## Computational analyses methods

**Gene Ontology (GO) enrichments.** For PPI GO enrichments, the human interacting proteins of each IAV protein were collated across all strains and cell types, and tested for enrichment of GO Molecular Function terms. The over-representation analysis was performed using the enrichGO function of clusterProfiler package (version 3.18.0) in R with default parameters. GO terms were obtained from the R annotation package org.Hs.eg.db (version 3.12.0). Significant GO terms were defined as those with p-value < 0.002. Terms with overlapping genes in each set were compared and the most significant term (lowest p-value) with the largest gene set size was selected as the non-redundant term. PPI enrichments were subject to further manual curation, with a maximum of the top three significant non-redundant GO terms listed and visualized for each IAV protein (Supplementary Data 1).

For PH GO enrichments, proteins with significantly up- and down-regulated phosphorylation events (defined as adjusted p-value < 0.05, absolute(log2FC) > 1, and observed in infected and mock samples) of each virus strain were collated at 18 hours post-infection (pH1N1, H3N2) and at 12 hours post-infection (H5N1) across all cell types, and tested for enrichment of GO terms from among all three ontologies: Biological Process, Molecular Function and Cellular Component. The over-representation analysis was performed using the enricher function of clusterProfiler package (version 3.12.0) in R with default parameters. GO terms were obtained from the R annotation package org.Hs.eg.db (version 3.12.0). Significant terms were defined as those with adjusted p-value < 0.05. We selected a set of non-redundant terms following an automated clustering procedure. We first constructed a term tree based on distances (1-Jaccard Similarity Coefficients of shared genes in KEGG or GO) between the significant terms. The term tree was cut at a specific level (h = 0.99) to identify clusters of non-redundant gene sets. For results with multiple significant terms belonging to the same cluster, we selected the most significant term (i.e. lowest adjusted p-value) (Supplementary Data 2).

**Network visualizations.** All networks were generated and visualized in Cytoscape (version 3.8.2)[129]. For the IAV-human PPI network, IAV-human PPIs were represented by strain and collated across all cell types. In cases where one human protein is shared between two virus proteins, the maximum MiST score from either IAV protein in any cell type was reported for each IAV strain. Human-human PPIs were annotated as reported in the comprehensive resource of mammalian protein complexes (CORUM) database[130]. Manual annotations to the network include human-human PPI protein complex name and biological process. Briefly, human prey proteins for each IAV protein were subjected to gene set enrichment analysis using either GO Biological Process terms or CORUM protein complex annotations. Genes that were members of enriched biological processes or protein complexes were labeled in the network using Adobe Illustrator software (v24.1). Labels were manually curated to simplify and generalize terms to facilitate interpretability. Genes mapped under a GO Biological Process term were manually investigated to ensure the term represented each gene's canonical function; genes that clearly possessed multiple functions, or genes that were otherwise difficult to classify, were excluded to reduce the appearance of misleading annotations. For the

PPI-phosphorylation overlay network, proteins were selected and visualized if they were identified in the PPI data above MiST scoring thresholds and in the phosphorylation data at the restricted time points (18 hours post-infection pH1N1, H3N2; 12 hours post-infection H5N1) with an absolute(log2FC) > 1 and adjusted *p*-value < 0.05 in any cell type. If a site was detected across multiple cell lines, the maximum absolute value, non-infinite fold-change was used.

**Kinase enrichment analysis.** Kinase activity predictions were generated on the full, non-infinite unfiltered log2FC phosphorylation data (i.e. no thresholds) using the ProtMapper resources available in OmniPath[71]. ProtMapper is a comprehensive catalog of kinase-substrate relationships that includes six databases (PhosphoSitePlus, SIGNOR, HPRD, NCI-PID, Reactome and the BEL Large Corpus) and three text-mining tools (REACH, Sparser and RLIMS-P). Kinase activity predictions were reported as a Z score that was calculated using the mean log2FC of phosphorylated substrates for each kinase in terms of standard error ($Z = [M - u] / SE$), comparing fold changes in phosphorylation site measurements of the known substrates against the overall distribution of fold changes across the sample. A *p*-value was also calculated by this approach using a two-tailed Z-test method, and thresholds were set at a FDR (by Benjamini-Hochberg method) <0.05. This approach has been previously shown to perform well at estimating kinase activities[19,79,131,132]. In our study, this approach produced annotations for 314 kinases with available data across all collected time points, virus strains and cell types (Supplementary Data 2). High-confidence kinase activity annotations were generated by: (1) limiting analysis to 18 hours post-infection (pH1N1, H3N2) and 12 hours post-infection (H5N1) in NHBE and THP-1 cell types, and (2) requiring each kinase possess two known phosphorylation sites detected in the PH dataset. This produced a high-confidence list of 13 kinases with activity predictions (Fig. 3F). Each site was represented individually or as a combination of phosphorylated sites when multiple phosphorylations were observed within single peptides. Kinase activities from SARS-CoV-2 phosphoproteomics data[19] were calculated using the same approach reported in this study, and also thresholded at FDR < 0.05 (Supplementary Fig. 6D).

**Influenza patient cohort analysis methods**
**Human subjects research.** We obtained de-identified human samples from 495 participants following written informed consent by the participants or their guardians at five eMERGE study sites (Cincinnati Children's Hospital Medical Center (CCHMC), Marshfield, Mount Sinai, Northwestern University, and Vanderbilt). Of the 495 participants, the mean age (with standard deviation) was 39.7 (+/−25) years (range: 0 to 90). 273 participants (55.2%) were female, and all were of European descent. We considered demographic information, including sex, race, ethnicity and age of the participants, as variables in the analyses. We used genetically determined sex information. We based the diagnosis of IAV infection on established clinical criteria. Approval for human subjects research was obtained from the institutions involved, and human subjects research was conducted in compliance with all relevant ethical regulations. Study protocols were reviewed and approved by the appropriate local institutional review boards (IRB) (Northwestern University IRB IDs STU00084534, STU00078215, STU00206610 and STU00211941).

**Whole-exome capture and DNA sequencing.** Exome enrichment was accomplished with the NimbleGen SeqCap EZ Exome+UTR (Roche NimbleGen, version 2) that targets 64 Mb of coding exons and miRNA regions plus 32 Mb untranslated regions (UTRs) for solution-based capture following the manufacturer's protocol. Library preparation was performed with 200 ng of genomic DNA using KAPA HyperPlus library kit (Roche, KK8514) using adaptors compatible with Illumina sequencer on the Hamilton STAR automated platform. We performed

amplification, pooling, hybridization, washing, and elution according to the manufacturer's instructions. We assessed the libraries for quality with a high sensitivity DNA ScreenTape assay on the 2200 TapeStation System (Agilent) and quantity with KAPA Library Quantification Kits for Illumina platforms (Kapa Biosystems). The libraries were diluted to 2 nM and clustered using an Illumina cBot with a HiSeq 3000/4000 paired-end cluster kit on a patterned flow cell and a HiSeq 3000/4000 SBS kit (300 cycles, Illumina v2.5 reagents) on the HiSeq 4000 sequencing platform.

**Data processing.** Each individual's WES data were mapped to the human reference genome (build hg19) using the Burrows-Wheeler Aligner (v0.7)[133]. After marking duplicates using Picard (http://broadinstitute.github.io/picard/), the Genome Analysis Toolkit (GATK, v3.1)[134] was used to remove duplicates, perform local realignment, and map quality score recalibration to produce a BAM file. Single nucleotide polymorphism (SNP) calls were made by the Haplotype-Caller (v3.4), filtering poor calls by the Variant Quality Score Recalibration (VQSR) filter from GATK. We sorted the aligned reads based on genome position using Picard (http://broadinstitute.github.io/picard/) and recalibrated the base quality score using default parameters. Each sample's final gvcf files were streamed to Illumina DRAGEN Bio-IT Platform (v3.1, Illumina), applying the default parameters in the '--vc-enable-gatk-acceleration true' option to identify and remove low-quality variants. We identified 3,621,267 genetic variants from the exome sequencing data after quality control, of which 3,256,844 had MAF < 1%. We performed quality control on our samples based on the number of total variants, number of singletons, missing rate, heterozygote vs homozygote ratio, transversion vs transition ratio, race inconsistency, sample relatedness, and missing phenotype. We finally identified 495 unrelated individuals of genetically-identified European Ancestry using PCA with LD-pruned variants (r2 = 0.2) and 1000 genome project phase3 data as ref. [135].

**Gene-based association and phosphorylation site prediction.** We classified a variant as pLOF if the variant was predicted deleterious (nonsynonymous exonic, frameshift substitution, or stop gain/loss variant; MAF < 1%) from any of the following six annotation algorithms: SIFT and SIFT 4G[75], PolyPhen-2 HDIV and PolyPhen-2 HVAR[76], likelihood ratio test (LRT), and Mutation Taster[77]. Variant annotations and pLOF were predicted by ANNOVAR using dbNSFP[136]. Using a gene-based collapsing method with various filters applied, we binned together the pLOF variants to identify their contributions to disease with good power. Firth logistic regression and burden tests[137] were applied, adjusting for hospital, age, sex, and top 10 principal components (PCs). FDR from Benjamini & Hochberg[138] < 0.05 was applied to identify genes as significant (Supplementary Data 3). Each base position was converted to codon coding using Ensembl Variant Effect Predictor (VEP)[139] and RefSeq as reference data. Kinase-specific phosphorylation site prediction was performed on the tested variants from the sets of genes involved (FDR < 0.05). Phospho-serine, phospho-threonine, and phospho-tyrosine sites were predicted using PhosphoSitePlus[140]. The prediction scores greater than and equal to 0.5 were considered predictive of a protein phosphorylation site. The rare variant discovery power for phosphorylation disrupting mutations was limited when not restricted to pLOF variants.

**IAV siRNA screen methods**
**siRNA reverse transfection.** A549 cells were reverse transfected in arrayed, 24-well format with 290 gene-targeting siRNA (Dharmacon, siGENOME siRNA SMARTpool cherry picked pre-designed library, 0.1 nmol/well), non-targeting control siRNA (Dharmacon, D-001206-14-05) or IAV NP-targeting control siRNA (Dharmacon, custom sequence 5'-GGAUCUUAUUUCUUCGGAGUU-3'). In each well of a 24-well plate (Fisher, 08-772-1), siRNA was diluted to final concentration

75 nM gene-targeting siRNA, 75 nM non-targeting siRNA, or 30 nM NP-targeting siRNA in 100 μL with OptiMem Reduced Serum Media (Thermo Fisher, 31985062). One NT and one NP siRNA per 24-well plate were included for each replicate. 2 μL/well Lipofectamine RNAiMAX Transfection Reagent (Thermo Fisher, 13778075) and 98 μL/well Opti-Mem media were mixed and incubated for 5 minutes. 100 μL RNAiMAX mix and 100 μL siRNA dilution were combined in each well, mixed and incubated for 20 minutes. During this incubation, A549 cells were trypsinized with 0.25% trypsin EDTA (Fisher, MT 25-053-CI), pelleted at 1200 rpm for 5 minutes, and resuspended in DMEM with L-glutamine without sodium pyruvate and 20% FBS at a density of $3 \times 10^5$ cells/mL. After the 20-minute incubation, 200 μL of $6 \times 10^4$ A549 cells were added to each well and returned to incubate at 37 °C and 5% CO2 for 48 hours. The experiment was performed in two sets for PPI and PH targets, each with two replicates per gene to assay IAV infectivity and one replicate per gene to assay cell viability.

**IAV infections.** All IAV infections were performed in accordance with BSL2* biosafety procedures. A549 cells were infected in 24-well format 48 hours after reverse transfection. Cell media was aspirated and cells were washed with 400 μL 1X PBS. Cells were infected at MOI 0.1 with Influenza A/WSN/1933 H1N1 virus strain (kindly provided by S. Chanda lab) diluted in a total of 100 μL 0.5% BSA in 1X PBS per well. Plates were returned to incubate for 1 hour at 37 °C and 5% CO2, and rocked by hand every 10 minutes during incubation. Following adsorption, virus inoculum was aspirated, and 400 μL DMEM (with L-glutamine without sodium pyruvate) with 0.1% FBS, 0.3% BSA, 0.5 μg/mL TPCK-treated trypsin and 1X Pen/Strep, was added to each well. Cells were returned to incubate at 37 °C and 5% CO2. At 24 hours post-infection, cells were trypsinized with 0.25% trypsin, moved to a 96-well U-bottom plate, pelleted at $800 \times g$ for 3 minutes, and fixed in 150 μL 1% formaldehyde in 1X PBS. Cells were stored at 4 °C until cell staining and flow cytometry.

**Cell staining and flow cytometry.** The percentage of A549 cells infected with IAV was quantified by immunostaining for IAV NP followed by flow cytometry in 96-well format on an Attune NxT flow cytometer (Thermo Fisher), as described above (Validation of IAV-Human PPIs M2-ATP6V1A and NEP-AHNAK in IAV-Infected Cells Methods).

Live-cell amine-reactive viability staining followed by flow cytometry was used to quantify non-viable A549 siRNA knockdown cells. 48 hours after reverse transfection, cells were trypsinized with 0.25% trypsin, neutralized with DMEM with L-glutamine without sodium pyruvate, 10% FBS and 1X Pen/Strep, and transferred from 24-well plates to 96-well U bottom plates. To remove and exchange buffers between incubations and washes, cells were pelleted in 96-well plates at $800 \times g$ for 3 minutes. Cells were pelleted and incubated in 100 μL/well 1:500 Ghost Dye Red 710 (Tonbo Biosciences, 13-0871-T100) in 1X PBS for 20 minutes at room temperature protected from light. Cells were pelleted and washed twice with 100 μL/well MACS buffer (PBS, 2 mM EDTA, 0.5% BSA; filtered through 500 mL EMD Millipore Steri-cup™ Sterile Vacuum Filter Units .22 μM PVDF (Fisher, SCGVU05RE)). Cells were pelleted and resuspended in 150 μL 1X PBS, and immediately analyzed in 96-well format on an Attune NxT flow cytometer (Thermo Fisher). Forward scatter voltage was set at 60 and side scatter voltage at 280 (R1, A549 singlet cells), and Ghost Dye Red 710 signal quantified by red laser 2 at voltage 260 (dead cells). 100 μL of cells were acquisitioned and all events recorded at 1000 μL/min.

Final cell gating (Supplementary Fig. 6A) and quantification for data analysis of %Ghost 710+ cells (percent dead cells) and %NP+ cells (percent IAV infectivity) was performed with FlowJo software (version 9.3.2). For cell viability, %Ghost710+ cells (percent dead cells) and %Ghost 710- cells (percent alive cells) for each siRNA knockdown are reported as calculated with FlowJo (Supplementary Data 4). For

IAV infectivity, singlet cell count and %NP+ cells for each experimental siRNA are reported as calculated with FlowJo (Supplementary Data 4). Singlet cell count was used as a readout for cell viability of A549 cells with siRNA knockdown and IAV infection. Singlet cell count and %NP+ cells for each experimental siRNA was normalized to the mean cell count or %NP+ cells of non-targeting (NT) control siRNA corresponding to each set of 24-well plates transfected, infected, collected and stained concurrently. Log2 fold changes in viability (singlet cell count experimental siRNA vs NT siRNA) and in percent IAV infection (%NP+ cells experimental siRNA vs NT siRNA) were calculated from these values (Supplementary Data 4). Data were thresholded on log2 fold change due to the small sample size of the screens.

**SARS-CoV-2 siRNA screen methods**

**siRNA transfections.** A549 cells stably expressing the ACE2 receptor (A549-ACE2) were kindly provided by O. Schwartz and were maintained at 37 °C, 5% CO2 in DMEM supplemented with 10% FBS, Pen/Strep and 10 μg/mL blasticidin S (Sigma, SBR00022). An siRNA library (Dharmacon, OnTargetPlus siRNA SMARTpool cherry picked pre-designed library, 2 nmol/well) of 54 target genes of interest, a non-targeting control and an ACE2-targeting control was used to transfect A549-ACE2 cells, previously seeded at a density of 6250 cells per well in a 384-well plate. Briefly, 0.1 μL of Lipofectamine RNAiMAX reagent and 4 pmoles of each siRNA pool were diluted in a final volume of 10 μL of OptiMEM. Following 5 minutes of incubation, 10 μL of the siRNA-lipid complexes were added to the cells, which were then incubated for 48 hours. Cells were then either infected with SARS-CoV-2 or left untreated for another 72 hours to determine cell viability using the CellTiter-Glo luminescent viability assay (Promega, G7570) according to the manufacturer's protocol. Luminescence was measured in a Tecan Infinity 2000 plate reader, and the percentage of metabolically active cells was calculated by normalizing the values to those obtained in untreated (100% viability) and 4% formalin-treated (0% viability) conditions included in each experiment. Experiments were performed in technical triplicate, with two biological replicates for PPI targets (total $n = 6$ per gene) and three biological replicates for PH targets (total $n = 9$ per gene).

**Virus infections and qRT-PCR quantification.** The SARS-CoV-2 (BetaCoV/France/IDF0372/2020) strain was a kind gift from the National Reference Centre for Respiratory Viruses at Institut Pasteur Paris, and was propagated once in VeroE6 cells to generate viral stock. 48 hours post-transfection, A549-ACE2 cells were infected with SARS-CoV-2 at MOI 0.1 PFU per cell. Briefly, cell media was removed and 20 μL of viral inoculum, prepared in serum-free media, was added to each well. After 1 hour adsorption at 37 °C, the inoculum was removed and replaced by DMEM supplemented with 2% FBS and Pen/Strep. The supernatant was harvested 72 hours post-infection and heat-inactivated at 95 °C for 5 minutes. The presence of viral genomes was subsequently quantified using the Luna Universal One-Step RT-qPCR kit (New England Biolabs, E3005S). Specific primers targeting the N gene (5'-TAATCAGACAAGGAACTGATTA-3' [forward] and 5'-CGAAGGTGTGACTTCCATG-3' [reverse]) were used as previously described[141]. RT-qPCR was performed under the following cycling conditions in an Applied Biosystems QuantStudio 6 thermocycler: 55 °C for 10 minutes, 95 °C for 1 minute, 40 cycles of 95 °C for 10 seconds, followed by 60 °C for 1 min. The number of viral genomes in the supernatant was calculated by performing a standard curve with RNA derived from a viral stock with a known viral titer, and is expressed as PFU equivalents per mL. These data were then used to compute log2 fold changes for experimental siRNA normalized to replicate-matched non-targeting controls. The log2 fold changes were computed separately for each replicate, and the median and median absolute deviation (MAD) were then calculated for each sample across all its

replicates in the screen set (six replicates for PPI and nine replicates for PH). The PPI and PH screen sets were analyzed separately, and the results are reported in Fig. 5F–G, Supplementary Fig. 6B and Supplementary Data 4. *P*-values were calculated using two-sided Wilcoxon signed-rank tests. Data were thresholded on log2 fold change due to the small sample size of the screens.

### Antiviral compound screen methods

**Compound treatment, cytotoxicity and IAV antiviral assays.** PPI-targeting and kinase-targeting compounds were manually curated by target-specific literature search performed by specialists within our group. IAV PB2-targeting compound Pimodivir (VX-787) was included as a positive control due to its antiviral activity against multiple H1N1-, H3N2- and H5N1-subtype IAV in human cell and mouse models and in patients[142-144]. Compounds were purchased from vendors specified in Supplementary Data 5. Drug antiviral assays were performed as compounds and IAV strains were received and available, and each set of compounds was performed alongside the Pimodivir control. For drug antiviral assays, A549 cells were seeded at 8,000 cells per well in DMEM growth media described above in 96-well plates (Falcon, 353072) 24 hours before IAV infection. Cells were pretreated with compound 2 hours before infection, where cell growth media for the corresponding well was replaced with media containing the indicated concentrations of compound (Supplementary Figure 7), or the equivalent volume of DMSO vehicle (control). The only exception is Pimodivir, which was added at 0.2 μM, 0.06 μM, 0.02 μM, 7 nM, 2 nM or 0.8 nM. Each compound or DMSO vehicle was tested in triplicate. Cells were mock-infected for cell toxicity assay, or infected with IAV for antiviral assay. Drug-containing media was removed and replaced with A/California/04/2009 H1N1 (MOI 0.5 PFU per cell), A/Wyoming/03/2003 H3N2 (MOI 0.5 PFU per cell), A/Vietnam/1203/2004 H5N1 HALo (MOI 0.05 PFU per cell), or no virus in 0.5% BSA in 1X PBS containing TPCK-trypsin. Cells were incubated for 1 hour at 37 °C to allow virus adsorption. Virus inoculum was subsequently removed, and 100 μL of drug- or vehicle-containing media was added. Uninfected A549 cells were assayed for cytotoxicity in parallel with the antiviral assay, matched for time and concentration. For cytotoxicity, 10 μL of Cell Proliferation Kit I (MTT) labeling reagent (Roche, 11465007001) was added to each well to a final concentration 0.5 mg/mL, and incubated for 3 hours at 37 °C. 100 μL of solubilization solution (Roche, 11465007001) was then added to each well, and plates were incubated at 37 °C overnight. Spectrophotometrical absorbance of each well was measured using a microplate (ELISA) reader (BioTek Instruments, NEO2SM) to quantify cell viability. For IAV infection, cells were fixed to the 96-well plate in final concentration 4% formaldehyde for 20 minutes and immunostained for IAV NP protein with a DAPI counterstain at room temperature. Briefly, cells were washed three times with 1X PBS for 5 minutes, permeabilized with 0.1% Triton X-100 (Fisher Scientific, 9002-93-1) in 1X PBS for 15 minutes, blocked in 1% BSA in 1X PBS for 1 hour and incubated in 1:1000 anti-IAV NP (an in-house monoclonal antibody HT103, provided by Dr. Thomas Moran, Thomas.Moran@mssm.edu) and DAPI (Thermo Scientific, 62248) for 1 hour. Cells were washed again three times with 1X PBS for 5 minutes, and incubated in 1:1000 goat anti-mouse AlexaFluor 488 (Invitrogen, A11029) in the dark for 1 hour. Cells were washed twice in 1X PBS for 5 minutes, suspended in 1X PBS and subsequently analyzed by Celigo Image Cytometer (Nexcelom) using instrument cell counting software to count the total number of IAV-infected cells (green channel, 536 nm). Infectivity was measured by the accumulation of viral NP protein (fluorescence accumulation). Percent infection was quantified as ((Infected cells/Total cells) - Background)*100, and the DMSO control was then set to 100% infection for analysis. Data analysis was performed in GraphPad Prism (version 9.3.0), using nonlinear regression fit and fit hill functions to identify IC50, IC90, CC10 and CC50 values (Supplementary Data 5). Selectivity index (SI) for each compound was calculated as CC50/IC50 (Supplementary Data 5), and compounds with a SI > 2 were reported as antiviral.

### Reporting summary

Further information on research design is available in the Nature Portfolio Reporting Summary linked to this article.

## Data availability

PPI, global AB, global PH and targeted (PRM) mass spectrometry data generated in this study have been deposited to the ProteomeXchange Consortium via the PRIDE[145] partner repository with the following dataset identifiers: 1) PPI data: PXD036077; 2) AB and PH data: PXD035900; and 3) PRM data: PXD041663. WES genotype data have been deposited to dbGaP and are available with the following accession: phs003407.v1.p1. Supplementary Data 1-5 provide data for graphs and/or full analyses for proteomic (PPI, AB, PH and PRM) data, pLOF data, siRNA screens and drug screens. IAV protein expression vectors are available from the authors upon request. Source data are provided with this paper.

## Code availability

R package source materials for MiST and MSstats are available through the Krogan Lab Github (https://github.com/kroganlab). Permanent reference versions used in this study are available for MiST (version 1.0.1) (https://doi.org/10.5281/zenodo.8034496)[146] and our slightly customized version of MSstats (version 3.99) (https://doi.org/10.5281/zenodo.8035059)[147]. All other sources of code for computational analyses were derived from publicly available websites and previous publications, and are cited in the corresponding Methods sections.

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

## Acknowledgements

This work represents a highly collaborative, multi-institutional research
endeavor, and we would like to thank and acknowledge members of the
National Institutes of Health (NIH) National Institute of Allergy and
Infectious Diseases (NIAID) U19 FluOMICS team and the University of
California San Francisco (UCSF) Quantitative Biosciences Institute (QBI)
Coronavirus Research Group (QCRG). This research was funded by NIH
NIAID U19 FluOMICS (AI106754) and FluOMICS Next Gen (AI135972)
awarded to E.-Y.K., L.P., M.L.S., S.K.C., R.A.A., S.M.W., A.G.-S. and N.J.K.
This work was also partially supported by the following awards to A.G.-
S.: by the Center for Research on Influenza Pathogenesis and Trans-
mission (CRIPT), an NIAID-funded Center of Excellence for Influenza
Research and Response (CEIRR, contract number 75N93021C00014); by
NIH NIAID grant U19AI142733; by the Department of Defense grant
W81XWH-20-1-0270; and by the Defense Advanced Research Projects
Agency (DARPA) grant HR0011-19-2-319-0020. In addition, this work was
supported by the following awards to N.J.K.: by U19AI135990 from the
NIH; by the Excellence in Research Award (ERA) from the Laboratory for
Genomics Research (LGR), a collaboration between UCSF, University of
California Berkeley, and GSK (#133122 P); by a Fast Grant for COVID-19
from the Emergent Ventures program at the Mercatus Center of George
Mason University; by the Roddenberry Foundation; by funding from F.
Hoffmann-La Roche and Vir Biotechnology and gifts from QCRG phi-
lanthropic donors; and by DARPA under Cooperative Agreement
#HR0011-19-2-0020. The views, opinions, and/or findings contained in
this material are those of the authors and should not be interpreted as
representing the official views or policies of the Department of Defense
or the U.S. Government. This work was also partially supported by
Electronic Medical Records and Genomics (eMERGE) grant U01HG011181
awarded to D.M.R., T.C.C., L.C.K., R.L.C., S.J.S. and S.M.W., and by the
following awards to S.M.W.: by NIH National Heart, Lung, and Blood
Institute (NHLBI) grant U01 HL146240, a Northwestern CORE Clinical
Research Site: Trans-omics for HIV/AIDS Research; and by NIH NIAID U01
AI035039, a Multicenter AIDS Cohort Study. Other funding support
includes NIH grant F32CA239333 awarded to M.B., and R21AI161104
awarded to C.F.B. K.M.H. was supported by the National Science
Foundation Graduate Research Fellowship Program (1650113). We thank
Dr. Anke Meyer-Franke and the Assay Development and Drug Discovery
Core at Gladstone Institutes for assistance with high-throughput
imaging. We also thank Laura Satkamp for support during cloning, and
Joseph Hiatt and Paige Haas for advice on flow cytometry analysis.

## Author contributions

Conceptualization: N.J.K., A.G.-S., R.M.K., K.M.H., J.F.H., B.K.S., T.A.T.,
J.E.M., K.M.S., S.M.W.; Methodology: K.M.H., M.J.M., A.F., R.M.K., J.F.H.,
V.V.R., A.H., E.S., S.P.; Software: B.J.P., Y.Z., M.B., H.B., B.H., E.-Y.K., H.K.;
Validation: K.M.H., M.J.M., L.M.-S., J.C.-S., A.S.J.; Formal Analysis: B.J.P.,
Y.Z., M.B., H.B., K.M.H., M.J.M, A.F., R.M.K., D.L.S, E.-Y.K., H.K., M.E.,
T.T.N., J.R.J., L.S.; Investigation: K.M.H., M.J.M., E.J.S., B.W.N., V.V.R.,
A.H., A.F., M.U., M.A.P.W., R.A.A., E.M., E.S., S.P., G.M., R.M.K., J.R.J.,
S.J.S., L.C.K., T.C.C.; Resources: M.S., T.A.T., B.K.S., R.A.A., S.K.C., L.M.-
S., J.C.-S.; Data Curation: K.M.H., R.M.K., D.L.S., L.P., E.-Y.K.; Writing -
Original Draft: K.M.H., R.M.K., L.Z.-A.; Writing - Review & Editing: K.M.H.,
N.J.K., R.M.K., M.E., L.Z.-A., M.B., J.B., J.F.H., R.A.A., A.G.-S., E.-Y.K., S.J.S.,
L.C.K., T.C.C.; Visualization: K.M.H., R.M.K., M.E., L.Z.-A., B.J.P., Y.Z., M.B.,
A.F., B.H., T.K.; Supervision: N.J.K., A.G.-S., R.M.K., J.F.H., D.L.S., J.R.J.,
B.K.S., K.M.S., M.V., M.S., K.M.W., S.M.W., S.K.C., C.F.B., D.R., R.L.C.,
J.A.P, M.E.S. Project Administration: N.J.K., A.G.-S.; Funding Acquisition:
N.J.K., A.G.-S., S.K.C., S.M.W.

## Competing interests

The Krogan Laboratory has received research support from Vir Bio-
technology, F. Hoffmann-La Roche, and Rezo Therapeutics. N.J.K. has
previously held financially compensated consulting agreements with
the Icahn School of Medicine at Mount Sinai, New York and Twist
Bioscience Corp. He currently has financially compensated consulting
agreements with Maze Therapeutics, Interline Therapeutics, Rezo Ther-
apeutics, and GEn1E Lifesciences, Inc. He is on the Board of Directors of
Rezo Therapeutics and is a shareholder in Tenaya Therapeutics, Maze
Therapeutics, Rezo Therapeutics, and Interline Therapeutics. The García-
Sastre laboratory has received research support from Pfizer, Senhwa
Biosciences, Kenall Manufacturing, Avimex, Johnson & Johnson, Dyna-
vax, 7Hills Pharma, Pharmamar, ImmunityBio, Accurius, Nanocomposix,
Hexamer, N-fold LLC, Model Medicines, Atea Pharma, Applied Biological
Laboratories and Merck, outside of the reported work. A.G.-S. has con-
sulting agreements for the following companies involving cash and/or
stock: Vivaldi Biosciences, Contrafect, 7Hills Pharma, Avimex, Vaxalto,
Pagoda, Accurius, Esperovax, Farmak, Applied Biological Laboratories,
Pharmamar, Paratus, CureLab Oncology, CureLab Veterinary, Synairgen
and Pfizer, outside of the reported work. A.G.-S. has been an invited
speaker in meeting events organized by Seqirus, Janssen, Abbott and
Astrazeneca. A.G.-S. is an inventor on patents and patent applications on
the use of antivirals and vaccines for the treatment and prevention of
virus infections and cancer, owned by the Icahn School of Medicine at
Mount Sinai, New York, outside of the reported work. The Hultquist
laboratory has received prior funding support from Gilead Sciences, and
J.F.H. has a financially compensated consulting agreement with Merck.
D.L.S. has a consulting agreement with Maze Therapeutics. M.B. is a
financially compensated scientific advisor for GEn1E Life Sciences.
K.M.S. has consulting agreements for the following companies involving
cash and/or stock compensation: Black Diamond Therapeutics,
BridGene Biosciences, Denali Therapeutics, Dice Molecules, eFFECTOR
Therapeutics, Erasca, Genentech/Roche, Kumquat Biosciences, Kura
Oncology, Merck, Mitokinin, Petra Pharma, Rezo Therapeutics, Revolu-
tion Medicines, Type6 Therapeutics, Vevo, Vicinitas and Wellspring
Biosciences (Araxes Pharma). J.C.-S. is a former employee and stock-
holder of Synthego. While not directly relevant to this study, C.F.B. is a
scientific advisor for Axion BioSystems, M.L.S. has a financially com-
pensated consulting agreement with Calibr, and S.P. is employed by
Roivant Sciences, Inc. S.P. also holds stock compensation from Roivant
Sciences and AbbVie. The remaining authors declare no competing
interests.

## Additional information

Kelsey M. Haas[1,2,3,4], Michael J. McGregor[1,2,3,4], Mehdi Bouhaddou[1,2,3,4], Benjamin J. Polacco[2,3,4], Eun-Young Kim[5], Thong T. Nguyen[1], Billy W. Newton[2,3], Matthew Urbanowski[6], Heejin Kim[5], Michael A. P. Williams[4,6,7], Veronica V. Rezelj[4,8], Alexandra Hardy[8], Andrea Fossati[1,2,3,4], Erica J. Stevenson[1,2,3,4], Ellie Sukerman[9], Tiffany Kim[5], Sudhir Penugonda[5], Elena Moreno[6,7,33,34], Hannes Braberg[2,3,4], Yuan Zhou[1,2,3,4], Giorgi Metreveli[6], Bhavya Harjai[1,2,3,4], Tia A. Tummino[3,4,10,11], James E. Melnyk[2,3,4], Margaret Soucheray[1,2,3,4], Jyoti Batra[1,2,3,4], Lars Pache[12], Laura Martin-Sancho[13,35], Jared Carlson-Stevermer[14,36], Alexander S. Jureka[15,16], Christopher F. Basler[6], Kevan M. Shokat[2,3,4,17], Brian K. Shoichet[3,4,10], Leah P. Shriver[18,19], Jeffrey R. Johnson[1,2,3,37], Megan L. Shaw[6,38], Sumit K. Chanda[13], Dan M. Roden[20,21,22], Tonia C. Carter[23], Leah C. Kottyan[24,25,26], Rex L. Chisholm[27], Jennifer A. Pacheco[27], Maureen E. Smith[27], Steven J. Schrodi[28], Randy A. Albrecht[6,7], Marco Vignuzzi[4,8], Lorena Zuliani-Alvarez[1,2,3,4], Danielle L. Swaney[1,2,3,4], Manon Eckhardt[1,2,3,4], Steven M. Wolinsky[5], Kris M. White[4,6,7], Judd F. Hultquist[4,5,29] ✉, Robyn M. Kaake[1,2,3,4] ✉, Adolfo García-Sastre[4,6,7,30,31,32] ✉ & Nevan J. Krogan[1,2,3,4] ✉

[1]J. David Gladstone Institutes, San Francisco, CA 94158, USA. [2]Department of Cellular and Molecular Pharmacology, University of California San Francisco, San Francisco, CA 94158, USA. [3]Quantitative Biosciences Institute (QBI), University of California San Francisco, San Francisco, CA 94158, USA. [4]Quantitative Biosciences Institute (QBI) Coronavirus Research Group (QCRG), San Francisco, CA 94158, USA. [5]Division of Infectious Diseases, Northwestern University Feinberg School of Medicine, Chicago, IL 60611, USA. [6]Department of Microbiology, Icahn School of Medicine at Mount Sinai, New York, NY 10029, USA. [7]Global Health and Emerging Pathogens Institute, Icahn School of Medicine at Mount Sinai, New York, NY 10029, USA. [8]Institut Pasteur, Viral Populations and Pathogenesis Unit, CNRS UMR 3569 Paris, France. [9]Division of Infectious Diseases, Oregon Health & Science University, Portland, OR 97239, USA. [10]Department of Pharmaceutical Chemistry, University of California San Francisco, San Francisco, CA 94158, USA. [11]Graduate Program in Pharmaceutical Sciences and Pharmacogenomics, University of California San Francisco, San Francisco, CA 94158, USA. [12]Infectious and Inflammatory Disease Center, Immunity and Pathogenesis Program, Sanford Burnham Prebys Medical Discovery Institute, La Jolla, CA 92037, USA. [13]Department of Immunology and Microbiology, The Scripps Research Institute, La Jolla, CA 92037, USA. [14]Synthego Corporation, Redwood City, CA 94063, USA. [15]Molecular Virology and Vaccine Team, Immunology and Pathogenesis Branch, Influenza Division, National Center for Immunization & Respiratory Diseases, Centers for Disease Control & Prevention, Atlanta, GA 30333, USA. [16]General Dynamics Information Technology, Federal Civilian Division, Atlanta, GA 30329, USA. [17]Howard Hughes Medical Institute, Chevy Chase, MD 20815, USA. [18]Department of Chemistry, Washington University in St. Louis, St. Louis, MO 63105, USA. [19]Center for Metabolomics and Isotope Tracing, Washington University in St. Louis, St. Louis, MO 63105, USA. [20]Department of Medicine, Vanderbilt University Medical Center, Nashville, TN 37232, USA. [21]Department of Pharmacology, Vanderbilt University Medical Center, Nashville, TN 37232, USA. [22]Department of Biomedical Informatics, Vanderbilt University Medical Center, Nashville, TN 37232, USA. [23]Center for Precision Medicine Research, Marshfield Clinic Research Institute, Marshfield, WI 54449, USA. [24]Center of Autoimmune Genomics and Etiology, Cincinnati Children's Hospital Medical Center, Cincinnati, OH 45229, USA. [25]Division of Human Genetics, Cincinnati Children's Hospital Medical Center, Cincinnati, OH 45229, USA. [26]Department of Pediatrics, University of Cincinnati College of Medicine, Cincinnati, OH 45229, USA. [27]Center for Genetic Medicine, Feinberg School of Medicine, Northwestern University, Chicago, IL 60611, USA. [28]Laboratory of Genetics, School of Medicine and Public Health, University of Wisconsin Madison, Madison, WI 53706, USA. [29]Center for Pathogen Genomics and Microbial Evolution, Northwestern University Havey Institute for Global Health, Chicago, IL 60611, USA. [30]Department of Medicine, Division of Infectious Diseases, Icahn School of Medicine at Mount Sinai, New York, NY 10029, USA. [31]Department of Pathology, Molecular and Cell-Based Medicine, Icahn School of Medicine at Mount Sinai, New York, NY 10029, USA. [32]The Tisch Cancer Institute, Icahn School of Medicine at Mount Sinai, New York, NY 10029, USA. [33]Present address: Department of Infectious Diseases, Hospital Universitario Ramón y Cajal and IRYCIS, Madrid, Spain. [34]Present address: Centro de Investigación en Red de Enfermedades Infecciosas (CIBERINFEC), Instituto de Salud Carlos III, Madrid, Spain. [35]Present address: Department of Infectious Disease, Imperial College London, London SW7 2BX, UK. [36]Present address: Serotiny Inc., South San Francisco, CA 94080, USA. [37]Present address: Department of Microbiology, Icahn School of Medicine at Mount Sinai, New York, NY 10029, USA. [38]Present address: Department of Medical Biosciences, University of the Western Cape, Bellville 7535 Western Cape, South Africa. ✉e-mail: judd.hultquist@northwestern.edu; robyn.kaake@ucsf.edu; adolfo.garcia-sastre@mssm.edu; nevan.krogan@ucsf.edu

