## [Peer Review File · Nature Communications]

nature portfolio

Peer Review FileReviewer comments, first round

Reviewer #1 (Remarks to the Author):

Influenza virus co-circulates with other respiratory viruses, including SARS-CoV-2, which caused the Covid pandemic. Accumulating data suggest that COVID vaccine might need to be updated annually as the flu vaccine. Thus, it is urgent to develop broad-spectrum antivirals. To address this question, Hass et al. adopted multiple omics approaches, such as AP-MS, phosphoproteomics, and exosome sequencing, to analyze influenza A virus-host interactions. They first mapped the IAV-host PPI network using 3 flu strains (H1N1, H3N2, and H5N1) in 3 different cell types (A549, THP-1 and NHBE). They identified 332 IAV-host PPIs and 44 of them are not found in previous IAV-host interactomes, such as the new NS2 interactor, AHNAK. Then, they performed global protein abundance and phosphorylation profiling on pH1N1, H3N2 or H5N1 IAV-infected primary NHBE and differentiated THP-1. They identified 13 kinases with activity changes during IAV infection, including 5 MAPKs. Furthermore, they performed exome sequencing of severe influenza patients and found 23 AB and 52 PH genes with pLOF variants to be significantly regulated during IAV infection. To examine the functional interactions, they adapted an arrayed siRNA screening approach in A549 cells and identified 54 host factors regulating IAV infection. To identify pan-viral host targets, they further examined the effects of knockdown of these 54 factors on SARS-CoV-2 infection and found that 3 of them acted as pan-respiratory virus regulators of IAV and SARS-CoV-2 infection. Lastly, to identify potential HDT against IAV infection, they screened compounds targeting a subset of the 44 siRNA-validated PPI factors and 13 kinases and found five compounds with pan-antiviral activity across IAV and SARS-CoV-2.

Overall, this is a well-designed systematic study and a well-written manuscript. The information derived from these omic studies will be a valuable resource to the field of influenza and coronavirus. Nonetheless, there are several minor suggestions that may make some improvements or clarifications.

1. It is known that the tag position could interfere with protein-protein interactions. The criteria for choosing either N or C tag will be useful information or guidelines for future protein interactome mapping. The author may show the data in the supplemental figures as it was marked as "data not shown" in the Methods.
2. In the Methods, the authors found that IFN treatment has minimal effects on PPI and combined the data into non-treatment groups. Although it might be unexpected, it might be better to discuss it in the main text.
3. Fig. S3. The PA-X expression in NHBE cells is pronounced; however, there is no WB signaling for GAPDH. It might need to be repeated.
4. The authors used a threshold log₂ fold change of <-2 or >2 for siRNA screening. The threshold is very stringent. However, the reason is not clear. For example, 2xSD or Z factor are commonly used. It might be due to the small sample size. The author may add discussion in the main text or Methods.

Reviewer #2 (Remarks to the Author):

The manuscript by Haas et al "Proteomic and genomic analyses of influenza A viruses identify pan-viral host targets" describes the results of a multi-pronged approach to identify druggable host targets of three influenza A virus strains. The authors combined co-precipitation-based analysis of protein-protein interactions by mass spectrometry, proteomics, phosphoproteomics, exome sequencing and siRNA knockdown and identified several target host proteins that can be targeted by FDA-approved drugs for potential combined treatment of flu and COVID-19. The paper is well written and the data collected from carefully designed experiments are impressive and novel and so are the conclusions. This will be an important paper for the fields of virology and omics and it will be interesting for many readers. My concerns, which are minor, are:

1. How do the doses used to show the efficacy of the compounds (figure 6) compare to the dosage

that was found safe for these drugs when they were approved for other treatments? (results of Phase 1 trials)?

2. When the authors write that the data for the three strains were collapsed, do they mean "collated"?

Reviewer #3 (Remarks to the Author):

In this manuscript, Haas and colleagues perform a systematic profiling of the global interactome, proteome, and phosphoproteome of three prototypic influenza strains, in three stereotypical cell lines commonly used in the field to study FluAV-host interactions to derive unique or redundant host targets of influenza. Furthermore, searching their data against a whole exome sequencing dataset from a large Influenza patients cohort, they leverage their experimentally-measured datasets to identify common targets with relevance in pathogenesis, and identify a specific mutation (S210P/Q) targeting a FluAV-modulated AHNAK proteoform (phosphoSer-210) as potentially associated with increased pathogenicity. To assess the functional relevance of newly or previously reported host targets identified by proteomics, the authors further employ an orthogonal approach, targeting 290 genes by siRNA screen in vitro, and a subset of these targets by pharmacological inhibition, to assess their drug-repurposing antiviral potential. Altogether, these experiments identify 3-5 (depending on the specific IAV strain used) novel inhibitors of IAV in vitro, and confirm other previously reported inhibitors including some associated with anti-IAV and anti-SARS-CoV-2 antiviral activity.

Overall, the study is a muscular tour-de-force both from a technical and a bioinformatics standpoint, and provides a remarkable example of integrative multi-omics. Data are clearly presented and results provided in light of the massive amount of literature existing for different Flu strains across domains (RNAi screen, PPI and Phosphoproteomes above all). Furthermore, the attempt to validate the data via pharmacological inhibition as well as genetic modulation is notable, and presents multiple interesting insights. I have only a few comments related to the extent of validation/biological follow-up of some of these large datasets (see below).

Major points:

As clarified above, the study already presents substantial orthogonal validation efforts (proteomics, patients cohorts, drug-repurposing, RNAi), however the observed effects in the RNAi screen and the criteria used to assess antiviral activities are rather loose/not stringent (RNAi: 4-fold difference cut-off, 1 siRNA/target, n=2 biological replicates; drug-repurposing: cytotoxicity cut-off not applied). Furthermore, none of the findings related to the proteomics datasets have been experimentally validated (either PPI or Phospho), including some of the new potentially interesting targets with relevance in pathogenesis (i.e. AHNAK).

Altogether, validating or following up on of these aspects would be paramount to consolidate the study and provide proof-of-principle of its main strength (using orthogonal multiomics to identify new targets).

- The study would greatly benefit from validation of at least some of the observed proteomics effects by alternative approaches (i.e. IF, WB) for the top prioritized proteins or at least some of the newly identified host targets.

For instance, are any of the newly identified PPI or PH or AB occurring during productive viral infection?

- One of the major strengths of the study (comparing the effects of viral orthologs across strains and cell lines to infer common and distinct mechanisms of host usage) is quickly overlooked by merging datasets for the three strains to prioritize candidate proteins. Indeed, there is a remarkable strain-specificity evident at different levels (i.e. the phosphoproteome being the most noteworthy; see Fig S5a). However, using these data to identify possible strain- or cell-specific effects might provide unique insights into differential pathogenesis outcomes. For instance, how does some of the differential susceptibility to drugs across strains (i.e. Lestaurtinib, pan-AKT, FLT3) relates to the species-specificities observed in the respective proteomics dataset? Are specificities observed at the proteomics level translated into spectrum of antiviral activity? It would be valuable to discuss some of the results in this light.

- Data presented by the authors suggest that phospho-null AHNAK mutants (S210P/Q) might be associated with more severe disease outcome; however, one of the most interesting observations of the study is not followed at all experimentally.

What is the impact of this mutation in vitro? Does phosphonull vs phosphomimetic variants of the protein differentially affect viral fitness? How this relates to circulating strains? (i.e. the AHNAK-NEP interaction was observed only for H5N1 in NHBE cells). Are H5N1 NEP mutants unable to bind this protein severely attenuated? What is their fitness in vivo?

Performing some of these experiments, even only the in vitro part, will ultimately strengthen the added value of such systematic, comprehensive and orthogonal approach.

- Was there any cut-off to cell viability applied in the drug-screen other than Selective Index calculation? Many of these compounds appear rather cytotoxic at concentrations exerting any antiviral activity. For instance Bafilomycin and Dinaciclib reduce cell viability of more than 25% already at the first concentration tested; some of the other compounds (i.e. Daunorubicin or Gilterinib) appear to exert antiviral activity only at cytotoxic concentrations.

Minor comments:

- While the use of bafilomycin A1 targeting ATP6V1A (M2 interactor) appears reasonable as "validation" control and proof-of-principle; the emphasis on its use of broad-spectrum HDT should be down toned significantly (inhibition of endocytosis affects nearly all enveloped viruses and is clearly a non-viable therapeutic avenue)

Reviewer #1 (Remarks to the Author):

Influenza virus co-circulates with other respiratory viruses, including SARS-CoV-2, which caused the Covid pandemic. Accumulating data suggest that COVID vaccine might need to be updated annually as the flu vaccine. Thus, it is urgent to develop broad-spectrum antivirals. To address this question, Hass et al. adopted multiple omics approaches, such as AP-MS, phosphoproteomics, and exosome sequencing, to analyze influenza A virus-host interactions. They first mapped the IAV-host PPI network using 3 flu strains (H1N1, H3N2, and H5N1) in 3 different cell types (A549, THP-1 and NHBE). They identified 332 IAV-host PPIs and 44 of them are not found in previous IAV-host interactomes, such as the new NS2 interactor, AHNAK. Then, they performed global protein abundance and phosphorylation profiling on pH1N1, H3N2 or H5N1 IAV-infected primary NHBE and differentiated THP-1. They identified 13 kinases with activity changes during IAV infection, including 5 MAPKs. Furthermore, they performed exome sequencing of severe influenza patients and found 23 AB and 52 PH genes with pLOF variants to be significantly regulated during IAV infection. To examine the functional interactions, they adapted an arrayed siRNA screening approach in A549 cells and identified 54 host factors regulating IAV infection. To identify pan-viral host targets, they further examined the effects of knockdown of these 54 factors on SARS-CoV-2 infection and found that 3 of them acted as pan-respiratory virus regulators of IAV and SARS-CoV-2 infection. Lastly, to identify potential HDT against IAV infection, they screened compounds targeting a subset of the 44 siRNA-validated PPI factors and 13 kinases and found five compounds with pan-antiviral activity across IAV and SARS-CoV-2.

Overall, this is a well-designed systematic study and a well-written manuscript. The information derived from these omic studies will be a valuable resource to the field of influenza and coronavirus. Nonetheless, there are several minor suggestions that may make some improvements or clarifications.

1. It is known that the tag position could interfere with protein-protein interactions. The criteria for choosing either N or C tag will be useful information or guidelines for future protein interactome mapping. The author may show the data in the supplemental figures as it was marked as “data not shown” in the Methods.

We thank the reviewer for pointing this out and have now addressed this in the text with the following explanation in the AP-MS Methods section: “The location for 2X-Strep tag insertion was informed by previously published studies. The 2X-Strep tag was inserted internally into the HA sequence at an insertion permissive site as previously described (<https://doi.org/10.1073/pnas.1320524110>). The 2X-Strep tag was cloned at the C-terminus of PB1, PB2, N40, PA-X and NP based on successful published functional studies conducted with these proteins tagged at the same position (doi: 10.1021/pr060432u; <https://doi.org/10.3389/fmicb.2019.00432>). For all other constructs, the 2X-Strep tag was cloned at the N-terminus, as this site was previously used to characterize the M1, M2, NS1 and NEP proteins (doi: 10.1016/j.bpj.2014.06.042; <https://doi.org/10.1038/s41467-019-12632-5>; <https://doi.org/10.1093/emboj/19.24.6751>), and N-terminal fusions are often used to generate recombinant NA (doi: 10.3390/v13101893).” We have removed “data not shown” in the Methods and replaced it with the citations.

2. In the Methods, the authors found that IFN treatment has minimal effects on PPI and combined the data into non-treatment groups. Although it might be unexpected, it might be better to discuss it in the main text.

We agree with the reviewer and have now included the following statement in the AP-MS Results section: “For each cell type after lentiviral transduction, three replicates were treated with interferon alpha to stimulate an antiviral-like state, and three replicates remained untreated. There were little discernible differences in

observed PPIs between interferon-treated and untreated samples, therefore samples were combined totaling six biological replicates (see **Methods** for details).”.

3. Fig. S3. The PA-X expression in NHBE cells is pronounced; however, there is no WB signaling for GAPDH. It might need to be repeated.

Thank you for pointing this out and we agree that having the GAPDH loading control blot to pair with the PA-X expression blot in NHBE cells would be helpful for interpretation. Therefore, we have repeated the WB using the original samples, and have updated Figure S3 accordingly. In making this correction, we also noticed our mistake in providing the WB for only one replicate for A549 cell M2 and PA samples, and have similarly repeated the WB using the original samples and updated Figure S3 accordingly. In our original manuscript draft, A549 cell M2 and PA samples were the only samples to use tubulin as a loading control; we updated Figure S3 to probe with GAPDH as a loading control to be consistent with all other samples and have removed tubulin from the figure, legend and Methods accordingly.

4. The authors used a threshold log₂ fold change of <-2 or >2 for siRNA screening. The threshold is very stringent. However, the reason is not clear. For example, 2xSD or Z factor are commonly used. It might be due to the small sample size. The author may add discussion in the main text or Methods.

We thank the reviewer for this comment. We chose to threshold based on log₂ fold change because it is calculated for each experimental sample relative to the mean or median of control samples within the screen, and was a more comparable reporting metric when utilizing multiple screens. The screens reported in our manuscript were completed at four different times, which is why we represent the data in four separate S-curves (i.e. PPI siRNA with IAV infection, PH siRNA with IAV infection, PPI siRNA with SARS-CoV-2 infection and PH siRNA with SARS-CoV-2 infection). Some of these screens were small in sample size (range: 10 genes to 212 genes), and we reasoned that using Z-scores or 2xSD could bias or skew the hits positively or negatively. The full RNAi screen data (no thresholds) are reported and available in the supplement, for readers to access and independently re-analyze by 2xSD cutoff if desired.

However, we agree our rationale was not made clear, therefore we have now added the following sentence in the IAV siRNA Screen Methods section and the SARS-CoV-2 siRNA Screen Methods section: “Data were thresholded on log₂ fold change due to the small sample size of the screens.”.

Reviewer #2 (Remarks to the Author):

The manuscript by Haas et al “Proteomic and genomic analyses of influenza A viruses identify pan-viral host targets” describes the results of a multi-pronged approach to identify druggable host targets of three influenza A virus strains. The authors combined co-precipitation-based analysis of protein-protein interactions by mass spectrometry, proteomics, phosphoproteomics, exome sequencing and siRNA knockdown and identified several target host proteins that can be targeted by FDA-approved drugs for potential combined treatment of flu and COVID-19. The paper is well written and the data collected from carefully designed experiments are impressive and novel and so are the conclusions. This will be an important paper for the fields of virology and omics and it will be interesting for many readers. My concerns, which are minor, are:

1. How do the doses used to show the efficacy of the compounds (figure 6) compare to the dosage that was found safe for these drugs when they were approved for other treatments? (results of Phase 1 trials)?

This is an excellent point and a question that we had considered. Many of the compounds we report are FDA-approved to treat multiple indications, or are in clinical trials for multiple indications, and to our knowledge, none of the existing indications for these compounds include infectious disease. Additionally, our studies are limited to cell culture models, in which dosages are not always directly comparative to in vivo models or clinical studies. For accuracy of reporting, we did not want to make comparisons between our study and clinical studies of non-infectious indications and/or of existing treatment paradigms that could be misconstrued as recommendations. We note if the compound we tested is in clinical trials (including phase) in Table S5, and encourage readers to investigate further if they are interested in assessing dosages used to treat individuals in these clinical studies.

2. When the authors write that the data for the three strains were collapsed, do they mean “collated”?

We apologize for the confusing terminology. We thank the reviewer for suggesting “collated” as it is a suitable choice; we meant we took the union of the datasets, collapsing against or de-identifying one variable (either cell type or strain). We have replaced the term “collapsed” with the term “unified” in all instances where it is used in the manuscript (Results, Figure Legends, Methods). We have additionally defined the term “unified” in the IAV PPI Networks Results section: “We therefore took the union of all PPIs across the three cell-type specific networks to generate one unified interactome to visualize pan-IAV and strain-specific interactions between 214 human proteins and 12 IAV proteins (Fig2)”.

Reviewer #3 (Remarks to the Author):

In this manuscript, Haas and colleagues perform a systematic profiling of the global interactome, proteome, and phosphoproteome of three prototypic influenza strains, in three stereotypical cell lines commonly used in the field to study FluAV-host interactions to derive unique or redundant host targets of influenza. Furthermore, searching their data against a whole exome sequencing dataset from a large Influenza patients cohort, they leverage their experimentally-measured datasets to identify common targets with relevance in pathogenesis, and identify a specific mutation (S210P/Q) targeting a FluAV-modulated AHNAK proteoform (phosphoSer-210) as potentially associated with increased pathogenicity. To assess the functional relevance of newly or previously reported host targets identified by proteomics, the authors further employ an orthogonal approach, targeting 290 genes by siRNA screen in vitro, and a subset of these targets by pharmacological inhibition, to assess their drug-repurposing antiviral potential. Altogether, these experiments identify 3-5 (depending on the specific IAV strain used) novel inhibitors of IAV in vitro, and confirm other previously reported inhibitors including some associated with anti-IAV and anti-SARS-CoV-2 antiviral activity.

Overall, the study is a muscular tour-de-force both from a technical and a bioinformatics standpoint, and provides a remarkable example of integrative multi-omics. Data are clearly presented and results provided in light of the massive amount of literature existing for different Flu strains across domains (RNAi screen, PPI and Phosphoproteomes above all). Furthermore, the attempt to validate the data via pharmacological inhibition as well as genetic modulation is notable, and presents multiple interesting insights. I have only a few comments related to the extent of validation/biological follow-up of some of these large datasets (see below).

Major points:

As clarified above, the study already presents substantial orthogonal validation efforts (proteomics, patients cohorts, drug-repurposing, RNAi), however the observed effects in the RNAi screen and the criteria used to assess antiviral activities are rather loose/not stringent (RNAi: 4-fold difference cut-off, 1 siRNA/target, n=2 biological replicates; drug-repurposing: cytotoxicity cut-off not applied). Furthermore, none of the findings related to the proteomics datasets have been experimentally validated (either PPI or Phospho), including some of the new potentially interesting targets with relevance in pathogenesis (i.e. AHNAK).

Altogether, validating or following up on of these aspects would be paramount to consolidate the study and provide proof-of-principle of its main strength (using orthogonal multiomics to identify new targets).

We thank the reviewer for their comments and agree that additional targeted validation will strengthen the manuscript and support our main premise of using a systems approach to identify new targets. We have therefore performed additional experiments (see comment below) to validate two targets, including AHNAK. While we agree that other studies have used more stringent cutoffs for various screens, we selected our cutoffs based on the size and design of the screens we used, and the desired outcome, which was a systems view of the data. In an effort to be as transparent as possible, we have reported all of our data from the full RNAi and compound screens without cutoffs, and with cytotoxicity reported (available in the supplement), for readers to access and independently re-analyze if desired.

As a minor point of clarification, each gene was targeted by a pool of 4 siRNA/target.

- The study would greatly benefit from validation of at least some of the observed proteomics effects by alternative approaches (i.e. IF, WB) for the top prioritized proteins or at least some of the newly identified host targets.

For instance, are any of the newly identified PPI or PH or AB occurring during productive viral infection?

We thank the reviewer for these comments. We would like to note that all of the abundance and phosphorylation proteomic data was collected during productive viral infection in two cell lines with three different strains over a time course of infection. We selected one time point that represented peak infection for downstream analyses presented in the manuscript; however, protein abundances and phosphorylation sites quantified at all other time points are reported in Table S2.

We agree that orthogonal lines of data validating IAV-human PPIs during infection could strengthen the manuscript. Accordingly, we have now included reciprocal IPs against endogenous host proteins in IAV-infected cells, and immunofluorescence (IF) staining of interacting host and IAV proteins in infected cells. First, we performed reciprocal IPs to validate IAV-human PPIs NEP-AHNAK and M2-ATP6V1A in infected cells. AHNAK was identified as an interactor of H5N1 IAV NEP in NHBE cells (SI Fig4, Table S1). ATP6V1A was identified as an interactor of M2 from all three IAV strains (pH1N1, H3N2 and H5N1) collated across the three cell types, and had the highest MiST interaction confidence score with H5N1 M2 in all three cell types (A549, THP-1 and NHBE) (SI Fig4, Table S1). We therefore focused on validating these interactions with A/Vietnam/1203/2004 H5N1 HALo IAV, which is the same H5N1 IAV strain used throughout our manuscript. We performed reciprocal IPs using antibodies against endogenous AHNAK (bait) in H5N1 IAV-infected NHBE cells, and endogenous ATP6V1A (bait) in H5N1 IAV-infected A549 cells. We used targeted mass spectrometry (MS) to identify and quantify abundance of prey IAV NEP and bait AHNAK proteins (NHBE cells), and prey IAV M2 and bait ATP6V1A proteins (A549 cells), in both input (cell lysate) and IP samples. Though we initially attempted to validate this interaction by reciprocal IP-WB, we could not reproduce commercial WBs with available AHNAK-targeting antibodies and thus decided to use quantitative mass spectrometry (Parallel Reaction Monitoring (PRM)) as our detection method instead. Using PRM, we quantified the enrichment of IAV NEP after pulling down with endogenous AHNAK or IgG1 control from H5N1-infected or mock-infected NHBE cells. In parallel, we performed similar enrichment/pull-down validation experiments for the M2-ATP6V1A interaction, which is a known functional host factor for IAV entry that had not yet been physically connected to IAV M2. As before, we used a PRM method to quantify the enrichment of IAV M2 after pulling down endogenous ATP6V1A or IgG control in H5N1-infected or mock-infected A549 cells. Additionally, as the ANHAK-NEP interaction is novel, we performed another complementary validation experiment for this interaction. Using IF staining against endogenous AHNAK and endogenous IAV NEP in NHBE cells infected with A/Vietnam/1203/2004 H5N1 HALo IAV, we observe that AHNAK and NEP both localize to the cytoplasm in infected NHBE cells.

We have now added all of these results to the main text and have updated **Fig S5** and the corresponding legend to incorporate all of the data. In addition, we have updated our methods section and submitted all the new proteomics data to the PRIDE repository (PXD041663 (Username: reviewer_pxd041663@ebi.ac.uk; Password: PwQzDzdF)).

- One of the major strengths of the study (comparing the effects of viral orthologs across strains and cell lines to infer common and distinct mechanisms of host usage) is quickly overlooked by merging datasets for the three strains to prioritize candidate proteins. Indeed, there is a remarkable strain-specificity evident at different levels (i.e. the phosphoproteome being the most noteworthy; see Fig S5a). However, using these data to identify possible strain- or cell-specific effects might provide unique insights into differential pathogenesis outcomes. For instance, how does some of the differential susceptibility to drugs across strains (i.e. Lestaurtinib, pan-AKT, FLT3) relates to the species-specificities observed in the respective proteomics dataset? Are specificities observed at the proteomics level translated into spectrum of antiviral activity? It would be valuable to discuss some of the results in this light.

We appreciate the reviewer pointing out many different strengths of our study, one of them being the opportunity to mine our data for host factors utilized in a strain-specific fashion. With the end-goal of exploring potential pan-viral HDT in mind, we focused our analyses and manuscript on highlighting pan-viral host factors that affect multiple strains of IAV, as well as an additional pandemic respiratory virus (i.e. SARS-CoV-2). We agree interesting examples of host proteins interacting with or changing at the proteomic level in a strain-specific manner exist in our dataset, and have explored some of these examples briefly in the Results section (e.g. PPI data discussed in text citing Fig2; phosphorylation data discussed in text citing Fig3F,G) and Supplemental Data (e.g. FigS4). Visualizing differences between the strains is also what motivated us to present side-by-side comparisons for pH1N1, H3N2 and H5N1 when representing PPIs and changes in AB or PH in our datasets (e.g. Fig2, Fig3, FigS4).

In response to the differential susceptibility to drugs across IAV strains, we do highlight one example (PRKDC) in the Host-Directed Compounds Results section. We have changed the language to more directly point out the comparison between the drug data and phosphoproteomic data in the text accordingly: “DNA-dependent protein kinase (DNA-PK) inhibitor NU7441 targeting M2 PPI DNA-PK PRKDC suppressed pH1N1 and H3N2 infection but not H5N1 infection (Fig6D). Interestingly, this strain specificity is also reflected in the phosphoproteomic data, as PRKDC showed increased kinase activity in pH1N1 and H3N2 infection but not H5N1 infection (Fig3F).”. To further touch on links between the drug data and phosphoproteomic data, we have included the following discussion about Lestaurtinib: “Lestaurtinib, which targets MAP2K3 and MAP2K6, showed antiviral activity against pH1N1 and H5N1 (SI > 2), and decreased H3N2 infection (SI < 2) (**Fig6H**). Interestingly, while MAP2K3 and MAP2K6 show increased predicted kinase activity in H3N2 and H5N1 IAV infection but not pH1N1 IAV infection (**Fig3F**), Lestaurtinib still inhibits pH1N1 IAV infection (**Fig6H**)”.

However, we acknowledge this does not provide a detailed comparative analysis. Given the depth, dimensionality and complexity of the data we report, it is not feasible to include all iterative comparisons within the scope of this manuscript. We hope our manuscript provides a reasonable “all encompassing” study, and we hope that bioinformaticians can download our datasets and perform these types of strain-specific comparative analyses for additional future bioinformatics-centric papers. We have now included this in our Discussion: “While our study focused on identifying promising antiviral targets for potential pan-viral HDT in future influenza and COVID-19 treatments, we recognize there is more to be mined from our data, especially in teasing apart strain- or cell-type specific interactions and their consequence on different disease prognoses or outcomes.”

- Data presented by the authors suggest that phospho-null AHNK mutants (S210P/Q) might be associated with more severe disease outcome; however, one of the most interesting observation of the study is not followed at all experimentally.

What is the impact of this mutation in vitro? Does phosphonull vs phosphomimetic variants of the protein differentially affect viral fitness? How this relates to circulating strains ? (i.e. the AHNK-NEP interaction was observed only for H5N1 in NHBE cells). Are H5N1 NEP mutants unable to bind this protein severely attenuated? What is their fitness in vivo? Performing some of these experiments, even only the in vitro part, will ultimately strengthen the added value of such systematic, comprehensive and orthogonal approach.

This is a great point and while we agree with the reviewer that these experiments would be very valuable, the results reported would likely be a fully publishable manuscript on their own. Currently, we feel they are beyond the scope of this paper, though we have plans to follow-up in the future.

- Was there any cut-off to cell viability applied in the drug-screen other than Selective Index calculation? Many of these compounds appear rather cytotoxic at concentrations exerting any antiviral activity. For instance Bafilomycin and Dinaciclib reduce cell viability of more than 25% already at the first concentration tested; some of the other compounds (i.e. Daunorubicin or Gilterinib) appear to exert antiviral activity only at cytotoxic concentrations.

Thank you for this comment. We took cell toxicity into account with the selectivity index (CC50/IC50), for which our cutoff was $SI \geq 2$. Beyond selectivity index, we did not introduce cutoffs for cell viability. We recognize the SI values are narrow for some of the compounds. However, SI is often dependent upon the cell line or model system. Our objective was to use the selectivity index cutoff to show that there is a separation between cytotoxicity and antiviral activity, and indicate which compounds are likely to have a true antiviral effect. We try to avoid indicating these compounds be used as therapies, but rather help point out that the host gene/protein itself is a promising antiviral target for future studies. To be as transparent as possible, we report full CC10, CC50, IC50 and IC90 data in Table S5 for readers to re-analyze and introduce different or additional cutoffs if desired. We have also reassessed the Introduction, Host-Directed Compounds and Discussion sections to ensure the language we use states these are promising antiviral targets (not therapies).

Minor comments:

- While the use of bafilomycin A1 targeting ATP6V1A (M2 interactor) appears reasonable as “validation” control and proof-of-principle; the emphasis on its use of broad-spectrum HDT should be down toned significantly (inhibition of endocytosis affects nearly all enveloped viruses and is clearly a non-viable therapeutic avenue)

We thank the reviewer for this point and agree with their comment. Accordingly we have removed ATP6V1A from the Discussion section about “broad-spectrum HDT”.

Reviewer comments, second round

Reviewer #1 (Remarks to the Author):

The comments I provided have been adequately addressed by the authors.

Reviewer #3 (Remarks to the Author):

I commend to the authors' efforts to strengthen the manuscript with additional experimental validation and I have no further comments.

Reviewer #1 (Remarks to the Author):

The comments I provided have been adequately addressed by the authors.

We thank the reviewer for their positive comment.

Reviewer #3 (Remarks to the Author):

I commend to the authors' efforts to strengthen the manuscript with additional experimental validation and I have no further comments.

We thank the reviewer for their positive comment.